# A glutamine-based single α-helix scaffold to target globular proteins

Albert Escobedo [1,9] ✉, Jonathan Piccirillo [1,10], Juan Aranda [1], Tammo Diercks [2], Borja Mateos [1], Carla Garcia-Cabau [1], Macarena Sánchez-Navarro [3], Busra Topal [1], Mateusz Biesaga [1], Lasse Staby [4], Birthe B. Kragelund [4], Jesús García [1], Oscar Millet [2], Modesto Orozco [1,5], Murray Coles [6], Ramon Crehuet [7] & Xavier Salvatella [1,8] ✉

The binding of intrinsically disordered proteins to globular ones can require the folding of motifs into α-helices. These interactions offer opportunities for therapeutic intervention but their modulation with small molecules is challenging because they bury large surfaces. Linear peptides that display the residues that are key for binding can be targeted to globular proteins when they form stable helices, which in most cases requires their chemical modification. Here we present rules to design peptides that fold into single α-helices by instead concatenating glutamine side chain to main chain hydrogen bonds recently discovered in polyglutamine helices. The resulting peptides are uncharged, contain only natural amino acids, and their sequences can be optimized to interact with specific targets. Our results provide design rules to obtain single α-helices for a wide range of applications in protein engineering and drug design.

Proteins are central to biology as they carry out a wide range of essential functions, from gene regulation to enzymatic catalysis, where their ability to specifically interact with other biomolecules is crucial. In pharmacology, inhibiting their interactions using drug-like small molecules is a common approach to modulate biological functions relevant to disease. In cases where the binding partner is another protein, the binding interfaces are usually flat and extended[1], making it challenging to inhibit the interactions with small molecules[2], and it is generally preferable to target them with antibodies. Yet, despite recent progress in intracellular antibody delivery[3], their clinical applications have been limited to targeting extracellular proteins, highlighting the

need to develop new molecular tools to inhibit intracellular protein-protein interactions.

Peptides have some of the advantages of small molecules, such as their ease of synthesis, and some of the advantages of antibodies, such as their relatively large size. Peptides, therefore, have at least in principle great potential as modulators of protein-protein interactions for pharmacological applications[4,5]. Protein-protein interactions where one partner is in a helical conformation are particularly common and especially amenable to inhibition by peptides: an excised linear peptide comprising a suitable sequence can, in principle, inhibit the interaction if it can bind to its partner with high affinity[6]. Linear

[1]Institute for Research in Biomedicine (IRB Barcelona), The Barcelona Institute of Science and Technology, Baldiri Reixac 10, 08028 Barcelona, Spain. [2]CIC bioGUNE, Basque Research and Technology Alliance (BRTA), Bizkaia Science and Technology Park, 48160 Derio, Spain. [3]Department of Molecular Biology, Instituto de Parasitología y Biomedicina López Neyra (IPBLN-CSIC), Armilla, Granada, Spain. [4]REPIN and Structural Biology and NMR Laboratory, The Linderstrøm-Lang Centre for Protein Science, Department of Biology, University of Copenhagen, Ole Maaloes Vej 5, DK-2200 Copenhagen N, Denmark. [5]Department of Biochemistry and Biomedicine, University of Barcelona, Avinguda Diagonal 645, 08028 Barcelona, Spain. [6]Department of Protein Evolution, Max Planck Institute for Biology, Max-Planck-Ring 5, 72076 Tubingen, Germany. [7]Institute for Advanced Chemistry of Catalonia (IQAC), CSIC, Jordi Girona 18-26, 08034 Barcelona, Spain. [8]ICREA, Passeig Lluís Companys 23, 08010 Barcelona, Spain. [9]Present address: Centre for Genomic Regulation (CRG), The Barcelona Institute of Science and Technology, Barcelona, Spain. [10]Present address: Department of Macromolecular Structures, Centro Nacional de Biotecnología (CNB-CSIC), Madrid, Spain. ✉e-mail: albert.escobedo@crg.eu; xavier.salvatella@irbbarcelona.org

peptides have a low propensity to fold into stable α-helices, however, and the entropic cost of folding decreases both their affinity for their targets and their stability against proteolytic degradation, highlighting the need to develop new tools to stabilize their helical conformation.

Introducing non-natural amino acids that act as potent N-caps[7] or substituting the $i+4 \rightarrow i$ hydrogen bonds stabilizing this secondary structure by covalent, and therefore permanent, surrogates[8] can be used to achieve this goal. Alternatively, specific amino acids can be introduced at positions $i$ and $i+3$, $i$ and $i+4$ or $i$ and $i+7$, that are close in space in α-helices, and linked by different means[9,10] such as by peptide stapling[11,12]. Peptide stapling is based on the use of pairs of synthetic α-methyl, α-alkenyl amino acids at relative positions $i,i+3$, $i,i+4$ or $i,i+7$ where their side chains can react by ring-closing metathesis when the peptide folds into an α-helix, thus greatly stabilizing this conformation[11,12]; in some cases, two such connections have been concatenated to obtain especially long and stable helices[13]. Although these approaches have shown applicability to inhibit protein-protein interactions, they have drawbacks that limit their range of applicability, such as their costly synthesis, limited solubility, and high rigidity.

We recently reported that Gln side chain to main chain hydrogen bonds stabilize the polyglutamine (polyQ) helix, a helical secondary structure formed by the polyQ tract of the androgen receptor (AR)[14,15]. In this structure both the main and side chain amide groups of Gln residues at position $i$ donate hydrogens to the main chain CO group of the residues at position $i$-$4$; the strength of this bifurcate interaction depends on the residue at position $i$-$4$, with Leu performing particularly well. Equivalent interactions have also been observed in the polyQ tract of the protein huntingtin, indicating that it is not specific to AR and can therefore be used in design[16]. As the relative donor and acceptor positions of such interactions are equivalent to those of the residues linked covalently in stapled peptides, we reasoned that side chain to main chain hydrogen bonds could be used to stabilize the α-helical conformation of linear peptides to enhance their interaction with globular proteins.

Here, to address this hypothesis we design a series of linear, uncharged, and highly soluble peptides and confirm their cooperative folding into single α-helices under physiological conditions. To evaluate the versatility and range of applicability of the design rules we analyze their tolerance towards changes of the acceptor residue, with bulky hydrophobic residues performing best, and introduce a pH-dependent conformational switch; in addition we explore how such bifurcated hydrogen bonds can be combined with electrostatic interactions between side chains to further stabilize α-helices. Most importantly, we show that the sequences of such peptides can be tailored to interact with specific globular proteins. In summary, the simple design rules that we propose can be used to engineer a class of linear, cooperatively folded helical peptides for use as templates in applications in pharmacology, materials science, synthetic biology and, more generally, in bioengineering.

## Results

### Design of Gln-based single α-helices

The helicity of peptides and intrinsically disordered (ID) proteins can be predicted by secondary structure prediction algorithms such as Agadir[17]. This algorithm, similar to the Zimm-Bragg[18] and Lifson-Roig[19] helix-coil transition models, is based on statistical mechanics. The statistical weight of the helical state of any peptide segment depends on the free energy of it folding into an α-helix, that Agadir computes as the sum of different terms including one accounting for interactions between residues at positions $i$ and $i+4$, close in space in α-helices, that requires experimental parametrization. The current version of Agadir does not account for the Gln to Leu ($Gln_{i+4} \rightarrow Leu_i$) side chain to main chain interactions stabilizing polyQ helices and therefore underestimates the helicity of the polyQ tract in AR:

peptide $L_4Q_{16}$, excised from AR and harboring four such interactions, has 38% helical propensity according to NMR experiments while Agadir predicts only 3%[14]. To address this we introduced an additional term to the free energy of folding into an α-helix accounting for this interaction ($\Delta G^{LQ}_{i,i+4}$) and by minimizing the prediction error with respect to the NMR-derived helicity (RMSD$_{Hel}$, see Supplementary Methods) obtained $\Delta G^{LQ}_{i,i+4} = -0.6$ kcal mol$^{-1}$ for $L_4Q_{16}$, in the range expected for one hydrogen bond in water[20] (Supplementary Fig. 1a). To our surprise we found that the value of $\Delta G^{LQ}_{i,i+4}$ that minimizes RMSD$_{Hel}$ depends on polyQ tract length: it increases from $-0.4$ kcal mol$^{-1}$ for $L_4Q_8$ to $-0.7$ kcal mol$^{-1}$ for $L_4Q_{20}$ (Supplementary Fig. 1b). The effective strength of the $Gln_{i+4} \rightarrow Leu_i$ interactions in AR peptides depends thus on the number of equivalent interactions following them in the sequence, suggesting cooperativity.

To analyze the origin of this behavior and exploit it for peptide design we studied four peptides of identical amino acid composition but with two potential $Gln_{i+4} \rightarrow Leu_i$ interactions (pink arrows in Fig. 1a) at different relative positions. The first such interaction is common to all peptides and the second one is shifted 1 (peptide P1-5), 2 (P2-6), 3 (P3-7), or 5 (P5-9) positions towards the C-terminus. After confirming that they were monomeric under our experimental conditions by size exclusion chromatography coupled to multiple angle light scattering (SEC-MALS) (Supplementary Fig. 1d), we used solution-state nuclear magnetic resonance (NMR) spectroscopy to probe their structural properties by exploiting the quantitative dependence of $^{13}C_\alpha$ and $^1H_\alpha$ NMR chemical shifts on residue-specific helical propensity, where larger $^{13}C_\alpha$ and lower $^1H_\alpha$ shifts indicate higher helicity[21]. An analysis of the NMR spectra indicated that in the sequence context of this family of peptides the strength of two $Gln_{i+4} \rightarrow Leu_i$ interactions is maximal when the donor of the first and the acceptor of the second share a peptide bond such that the two interactions are concatenated, as in peptide P3-7; these results were confirmed by circular dichroism (CD) spectroscopy (Supplementary Fig. 1c).

We then used CD spectroscopy to study the secondary structure of peptides containing two and three pairs of concatenated $Gln_{i+4} \rightarrow Leu_i$ interactions, $(P3-7)_2$ and $(P3-7)_3$, and obtained that they are highly helical and monomeric ($\Theta_{222 nm}/\Theta_{208 nm} < 1$) (Fig. 1c, d). An analysis of their secondary structure based on the main chain NMR chemical shifts indicated that the residue-specific helical propensity ($p_{Hel}$) is larger than 0.9 across 9 contiguous residues for $(P3-7)_2$ and larger than 0.95 across 16 residues for $(P3-7)_3$. We also characterized peptide $(P3-7)_{3 Ctrl}$, in which all Leu hydrogen bond acceptors were substituted by Ala, resulting, despite the higher helical propensity of Ala, in lower helicity (Fig. 1c, d) due to the relatively low stability of side chain to main chain hydrogen bonds accepted by Ala (see below)[14]. As it is not common for monomeric peptides to cooperatively fold into α-helices in the absence of tertiary interactions, we verified their monomeric state by SEC-MALS and native mass spectrometry (MS) (Supplementary Fig. 1d, e). In addition, we removed the N-terminal PGAS motif, which can facilitate helix nucleation[14], from these peptides (Fig. 1c) and observed that the *un*capped counterparts, $u(P3-7)_2$ and $u(P3-7)_3$, also have high helical propensity (Supplementary Fig. 1f).

Next, we investigated the thermal stability of the helices by CD spectroscopy at temperatures up to 368 K (Fig. 1e and Supplementary Fig. 1g). The spectra at 278 K were equivalent to those obtained upon cooling after thermal unfolding, indicating that the unfolded state of the peptide is soluble under our experimental conditions. We also characterized the structural properties of $(P3-7)_3$ by NMR at physiological temperature, 310 K. A comparison of the $^{13}C$-detected 2D CACO spectra of $(P3-7)_3$ at 278 and 310 K revealed only a small decrease in helical propensity (Fig. 1d, e) indicating that the peptide remains essentially fully folded at 310 K ($p_{Hel} \approx 0.90$ over 12 contiguous residues). Figure 1f shows the CD-monitored thermal denaturation of peptides $(P3-7)_3$ and $(P3-7)_{3 Ctrl}$, which reports on the higher stability of the former. Finally, prompted by the observation that these peptides

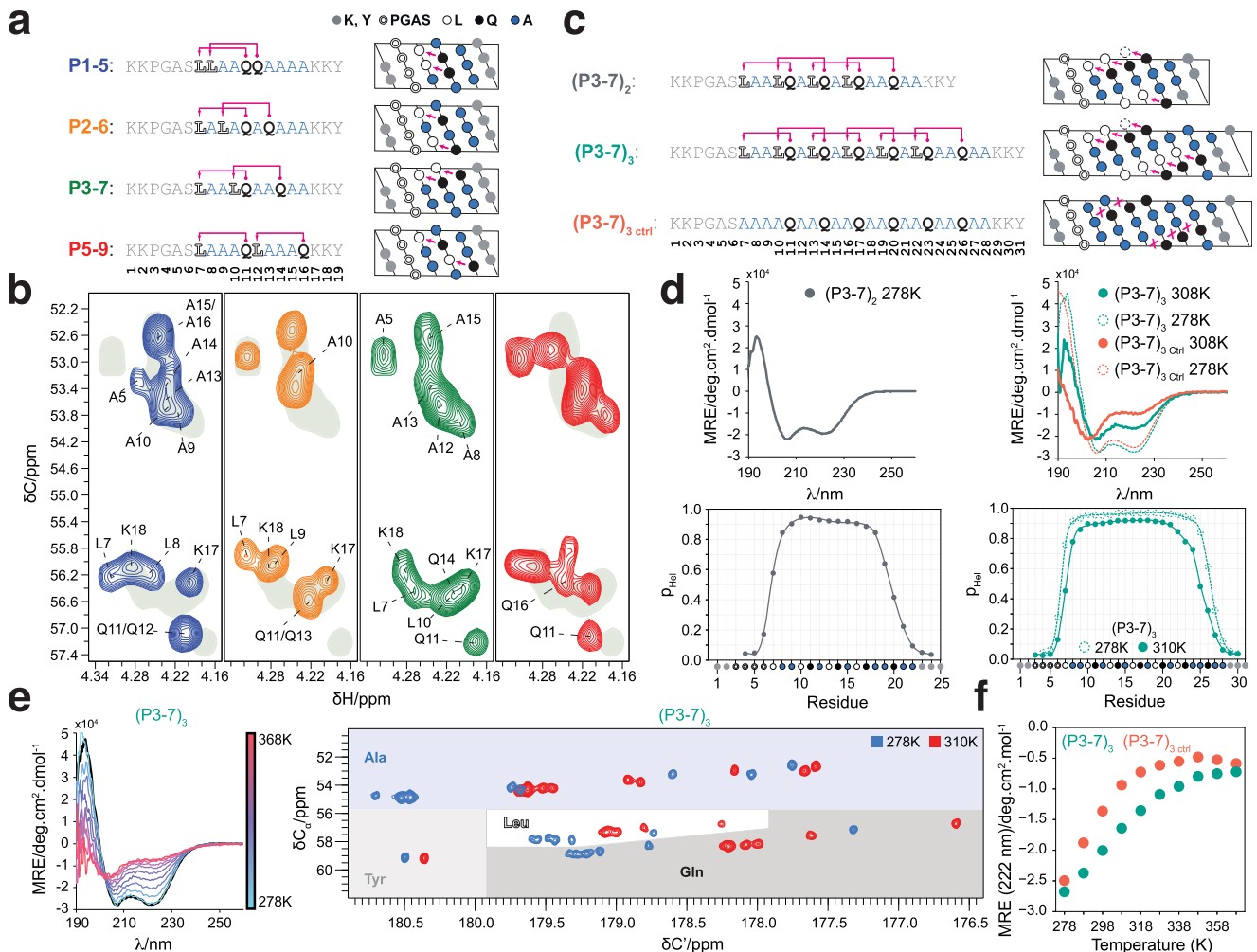

**Fig. 1 | Design of single α-helices stabilized by Gln side chain to main chain hydrogen bonds. a** Sequences and representation as helical projections of the peptides used to investigate cooperativity, where pink arrows indicate putative Gln side chain to main chain interactions. **b** Selected region of the 2D $^1$H$^{13}$,C HSQC spectra of the peptides at 278 K, where the shaded area corresponds to that of peptide P3-7, which has the highest helical propensity. **c** Sequence and representation as helical projections of peptides (P3-7)$_2$ and (P3-7)$_3$, as in **a**. **d** Top: CD spectra of peptides (P3-7)$_2$, (P3-7)$_3$, and (P3-7)$_{3\ Ctrl}$ at the indicated temperatures. Bottom: residue-specific helical propensities as obtained from the C′, C$_α$, N$^H$, and H$^N$ NMR chemical shifts by using CheSPI[65,66]. **e** Left: CD spectra of peptide (P3-7)$_3$ at temperatures between 278 K (blue) and 368 K (red) in steps of 10 K; the spectrum obtained at 278 K after refolding is shown in black. Right: superimposition of a selected region of the $^{13}$C-detected 2D CACO NMR spectra of peptide (P3-7)$_3$ recorded at 278 K (blue) and 310 K (red), with indications as colored shades of amino acid-specific regions. **f** Thermal denaturation of peptides (P3-7)$_3$ and (P3-7)$_{3\ Ctrl}$ monitored by measuring the mean residue ellipticity at 222 nm.

remain highly helical at physiological temperature, we also investigated their stability in human serum as well as their internalization in HeLa cells. Both (P3-7)$_3$ and (P3-7)$_{3\ Ctrl}$ show high stability in human serum, with half-lifes over 24 hours, well beyond that of Angiopep-2[22] that we included as a positive control (Supplementary Fig. 2a); in addition, both peptides were internalized by HeLa cells at 310 K (Supplementary Fig. 2b). In conclusion, concatenating Gln$_{i+4}$ → Leu$_i$ interactions allows obtaining stable α-helices that contain only natural amino acids, remain folded under physiological conditions, are soluble upon thermal unfolding, are resistant to proteolytic degradation, and are readily internalized by living human cells.

## Structure of a Gln-based single α-helix

The high quality of the NMR spectra obtained for peptide (P3-7)$_2$ allowed using this technique to study its structure at atomic resolution and further characterize the interactions stabilizing its helical conformation (Supplementary Fig. 3a). First, we measured $^{15}$N relaxation at 278 K (R$_1$, R$_2$, heteronuclear $^{15}$N{$^1$H} NOE) at two magnetic field strengths (14.1 T and 18.8 T) for the main chain amide (NH) groups of residues Gly4 to Lys24 and for the side chain amide (N$_{ε2}$H$_{ε21}$) groups of

all four Gln residues (Gln11, Gln14, Gln17, Gln20) (Fig. 2a and Supplementary Fig. 3b). We found that the main chain $^{15}$N R$_2$/R$_1$ ratios increase from the termini towards the center of the peptide until reaching plateau values (5.05 ± 0.25 at 18.8 T, 3.20 ± 0.15 at 14.1 T) between residues Leu 10 and Gln17 (Fig. 2a). A similar trend is traced out by the $^{15}$N{$^1$H} NOE, that reaches upper plateau values between 0.65 and 0.83 over a larger central segment, from Leu7 to Ala18. Although this result could be influenced by the well-characterized phenomenon of helix fraying[23] the main chain amide $^{15}$N relaxation data localizes the region of highest structural rigidity on a $10^{-1}$ to $10^1$ ns timescale as the one defined by the pairs of Leu and Gln residues involved in concatenated Gln$_{i+4}$ → Leu$_i$ interactions, i.e., from Leu 10 to Gln17; by contrast the Gln20 → Leu16 interaction, involving the last Gln residue, appears to be weaker.

The different properties of the first three Gln residues (11, 14, and 17) relative to the last Gln residue (20) are also evident in the side chain amide $^{15}$N$_{ε2}$ relaxation data: whereas the positive (at 18.8 T) or close to zero (at 14.1 T) $^{15}$N$_{ε2}${$^1$H$_{ε21}$} NOE values suggest significantly less side chain mobility for the first three Gln residues, Gln20 shows clearly negative values, suggesting higher side chain mobility (Fig. 2a), also

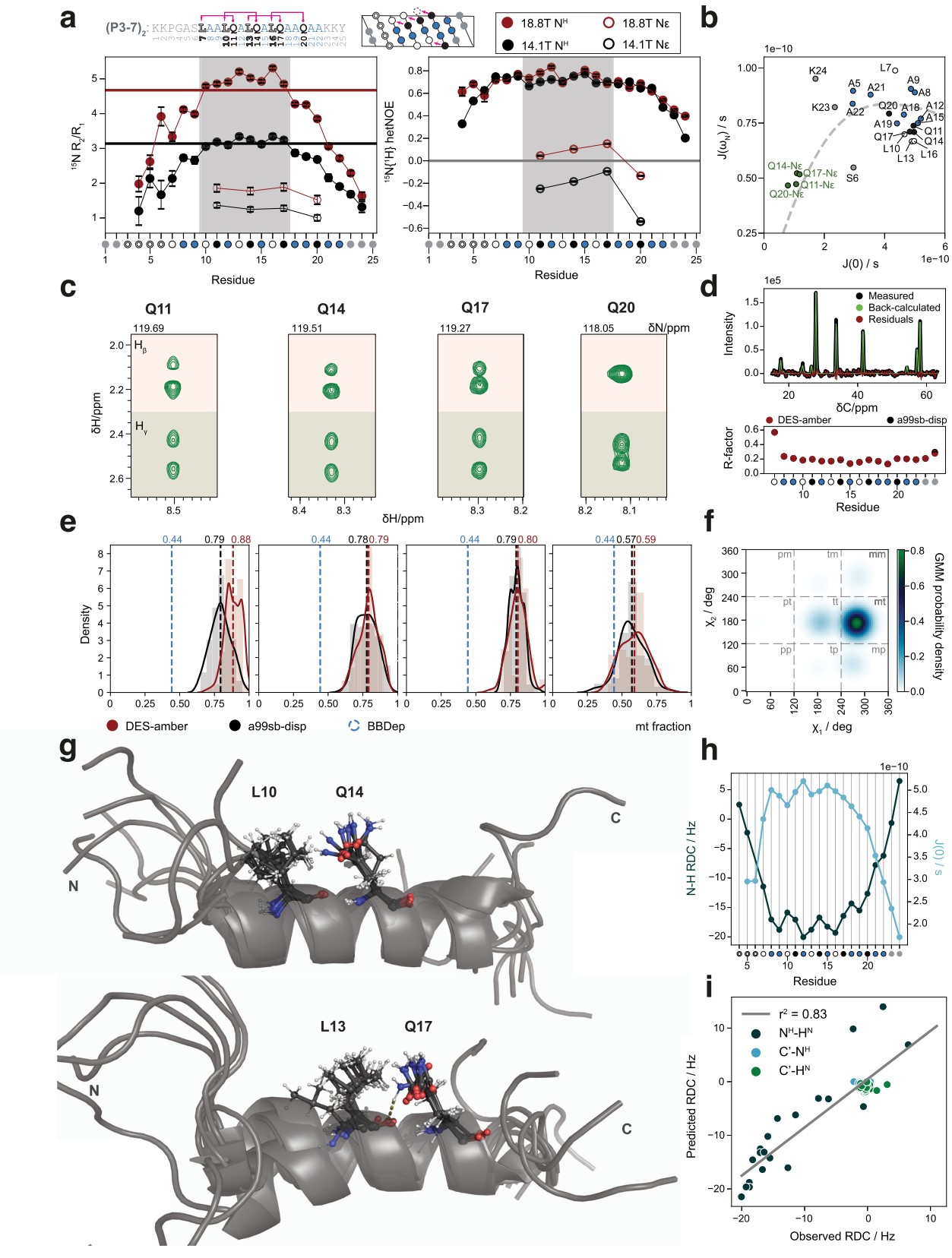

reflected in the relaxation-derived spectral density map (Fig. 2b). These results agree with the notion that the first three Gln residues form stronger $Gln_{i+4} \rightarrow Leu_i$ interactions than Gln20. The high NMR signal dispersion in both the $^{13}C$ and $^{15}N$ dimensions of the spectra of peptide $(P3\text{-}7)_2$ also allowed studying the Gln side chains: the first three Gln residues show fully resolved $H_\beta$ and $H_\gamma$ signals in the $^{15}N$-edited

HSQC-TOCSY spectrum, whereas the $H_\beta$ signals overlap for Gln 20 (Fig. 2c). This indicates that the conformations of the side chains of the former are better defined than those of the latter, even more than in polyQ helices[14,16], in agreement with the relaxation data.

To improve the description of the Gln side chains, we used CoMAND[24] to infer rotamer populations from a diagonal-free 3D

**Fig. 2 | Structure of a Gln-based single α-helix. a** $^{15}$N NMR relaxation data for main chain (N$^H$) and Gln side chain (Nε) amide groups of peptide (P3-7)$_2$, measured at 14.1 and 18.8 T at 278 K in 10% D$_2$O (main chain) and 50% D$_2$O (side chain). Left: $^{15}$N R$_2$/R$_1$ ratios, error bars represent the error-propagated SD of both exponential decay fits to R$_1$ and R$_2$ data and solid horizontal lines represent the theoretical R$_2$/R$_1$ ratio of a rigid body in isotropic motion with τ$_c$ = 4.5 ns, whose field dependency agrees well with the experimental data. Right: $^{15}$N{$^1$H} heteronuclear NOE, where the error bars correspond to the NOE SD[71] and the shade corresponds to the region with the highest degree of structuration. **b** Spectral densities derived by reduced spectral density mapping of all $^{15}$N relaxation data: the dashed line corresponds to the behavior expected for a polypeptide. **c** Strips from a 3D $^{15}$N-edited TOCSY-HSQC spectrum showing the H$_β$ and H$_γ$ resonances of all four Gln residues in (P3-7)$_2$. **d** Residue-specific CoMAND fitting of NOE signal intensities from the 3D CNH-

NOESY spectrum of (P3-7)$_2$. Top: example measured and back-calculated 1D $^{13}$C traces (Gln14 main chain N, a99sb-disp frame pool). Bottom: average R-factors obtained from 100 CoMAND iterations using either frame pool. **e** Density plots and mean values of the fraction of *mt* rotamer found in the 100 fitting iterations for all four Gln residues. The fraction of helical glutamines in *mt* configuration in the BBDep dataset[25] is shown in blue **f** Probability density for Gln14 χ$_1$ and χ$_2$ derived using the Gaussian mixture model (GMM). **g** Structural ensemble of (P3-7)$_2$, with main chain conformations selected by CoMAND from both trajectory pools and side chain conformations generated by GMM sampling. Top: one of 20 global ensemble calculations aligned for the Leu10-Gln14 pair. Bottom: same iteration aligned for the Leu13-Gln17 pair. **h** Residual dipolar couplings (RDCs) for the main chain $^1$H-$^{15}$N moieties and relaxation-derived main chain spectral density at zero frequency, J(0). **i** Correlation between experimental and back-calculated RDCs.

CNH-NOESY spectrum reporting on distances between protons bound to $^{15}$N and $^{13}$C. Briefly, CoMAND selects subsets of conformers from a pool, here obtained by molecular dynamics, that reproduce the NOESY spectra (Fig. 2d). The distributions were enriched in the *mt* Gln rotamer (χ$_1$ = −60° and χ$_2$ = 180°) that is required for the Gln$_{i+4}$ → Leu$_i$ interaction: the Gln residues involved in strong interactions (11, 14, and 17) have a high *mt* population (0.80), whereas that obtained for Gln 20 was lower (0.58). Both values are higher than that obtained for Gln residues in α-helices of structures deposited in the PDB[25], 0.44 (Fig. 2e); importantly, these results were robust to changes in the force field used to generate the pool[26,27].

To generate a conformational ensemble, we used the residue-specific CoMAND ensembles to train a Gaussian mixture model (GMM) by inferring χ$_1$ and χ$_2$ probability densities for each residue (Fig. 2f and Supplementary Fig. 3c), modified the side chain conformations accordingly and, through R-factor minimization, obtained the set of representative conformers shown in Fig. 2g (Supplementary Table 1). To validate the ensemble we measured three sets of residual dipolar couplings (RDCs) under steric alignment. The main chain $^1$D$_{H,N}$ values show a dipolar wave pattern, typical of α-helices, that matches well the period of the zero frequency spectral density, J(0) (Fig. 2h). Although the RDCs were not used as restraints they correlate well with the ensemble-averaged values (Q = 0.37, Fig. 2i), confirming that the ensemble is an accurate representation of peptide (P3-7)$_2$ and that the design rules that we have put forward lead to single α-helices.

## Ranking Gln$_{i+4}$ → X$_i$ interactions by strength

The stability of our Gln-based α-helices stems from concatenated Gln side chain to main chain hydrogen bonds accepted by Leu residues. This design decision was based on the fact that the polyQ tract found in AR, which is the most helical studied so far, is flanked by four Leu residues[14]. We sought to determine how other residues perform as acceptors to increase the versatility of the design rules and better understand the factors determining the strength of the interaction. For this, we used a host-guest approach in which we determined the secondary structure of L$_3$XQ$_{16}$ peptides (Supplementary Table 2) by NMR. These peptides were obtained by substituting the fourth Leu residue of peptide L$_4$Q$_{16}$, excised from AR, by 13 different representative amino acids (Fig. 3a and Supplementary Fig. 4a).

We measured the residue-specific helical propensities of the peptides by NMR by combining standard $^1$H$^N$-detected triple resonance with $^{13}$C-detected CACO and CON 2D NMR experiments: the high resolution in the CO dimension of the latter allowed the unambiguous assignment of all Gln residues, even in the variants with lowest signal dispersion (Supplementary Fig. 4b). Except L$_3$TQ$_{16}$ and L$_3$SQ$_{16}$ all variants show a helicity profile approximately proportional to that of L$_4$Q$_{16}$, (Fig. 3a): L$_3$TQ$_{16}$ and L$_3$SQ$_{16}$ instead show a different profile, likely because in these cases the substitution shifts the site of helix nucleation by introducing an S/T N-capping motif[28]. In these, Ser/Thr accept two concomitant hydrogen bonds donated by main chain amides of residues C-terminal to them: one by the

main chain O and another one by the side chain O of the hydroxyl group.

To explain the range of helicities obtained we hypothesized that it is due to two main factors: the intrinsic helical propensity of residue X[29] and, based on our previous work[14], the ability of its side chain to shield the hydrogen bond. While the former has been extensively measured[17,29,30], we quantified the latter by using molecular modeling, considering that the conformation of the residue accepting the hydrogen bond (X$_i$) can have an effect on the interaction of the Gln$_{i+4}$ H$_{ε21}$ donor with competing water molecules (Fig. 3b and Supplementary Fig. 7a). For this we computed 1 μs trajectories in different force fields[27,31] for all 20 possible variants in which we constrained the secondary structure of the Leu2-Gln5 segment (Supplementary Figs. 5a, 6a) and increased the population of the Gln$_{i+4}$ → Leu$_i$ interaction with a soft restraint (Supplementary Figs. 5b, 6b) to facilitate sampling the relevant region of conformational space: we obtained, as expected, that the higher the frequency of the hydrogen bond, the lower the solvent accessible surface area (SASA) of the H$_{ε21}$ atom of Gln 4, with high correlation (Fig. 3c).

Next, we quantified to what extent intrinsic helical propensity (x1) and Gln$_{i+4}$ H$_{ε21}$ SASA (x2) explain the experimental helical propensities (Fig. 3d) by multiple linear regression. Indeed we obtained that these two independent variables explain 73% of the variability (Fig. 3d). Remarkably, the Gln4 H$_{ε21}$ SASA value (x2) is the most important factor in the correlation, as its weight in the fitted equation is 40% higher than that of intrinsic helicity (x1); the model allows the prediction of the average helicities of the L$_3$XQ$_{16}$ variants not included in our experimental dataset (Fig. 3e). The correlation improves (from $r^2$ = 0.73 to 0.85) when only the subset of residue types with apolar side chains is considered, suggesting that additional factors might play a role when charged or polar side chains are present (Fig. 3f), and the results are robust to changes in MD force field[27,31] or intrinsic helical propensity scales (Fig. 3f). These data confirm that Leu is one of the best helicity-promoting acceptors, but that other residues such as Phe, Tyr, Ile or Met are similarly good, and that Trp is a particularly good acceptor in spite of its low intrinsic helical propensity. Thus, residues other than Leu can be introduced as acceptors of Gln side chain to main chain interactions, increasing the versatility of our design rules.

These results prompted us to investigate the presence of (P3-7)$_n$-like motifs in nature. To do this we searched UniprotKB[32], including the Swiss-Prot and TrEMBL databases, by using the motif search tool in ScanProsite[33], which we queried with the motif Ω-X-X-(Ω-Q-X)$_{n-1}$-X-Q-X-X, with Ω denoting good acceptors of Gln side chain to main chain hydrogen bonds (namely W, L, F, Y, I, M). We found that 3451 proteins contain sequences matching the motif, belonging to organisms across the kingdoms of life with representatives of a wide variety of taxonomic lineages including archaea, bacteria, viruses, and a full range of eukaryotes, from unicellular organisms to metazoa including humans (Fig. 3g). There is experimental evidence for the existence of 94 of these proteins (UniprotKB annotation score >3), mostly belonging to extensively characterized metazoa (Supplementary Fig. 7b). Figure 3h

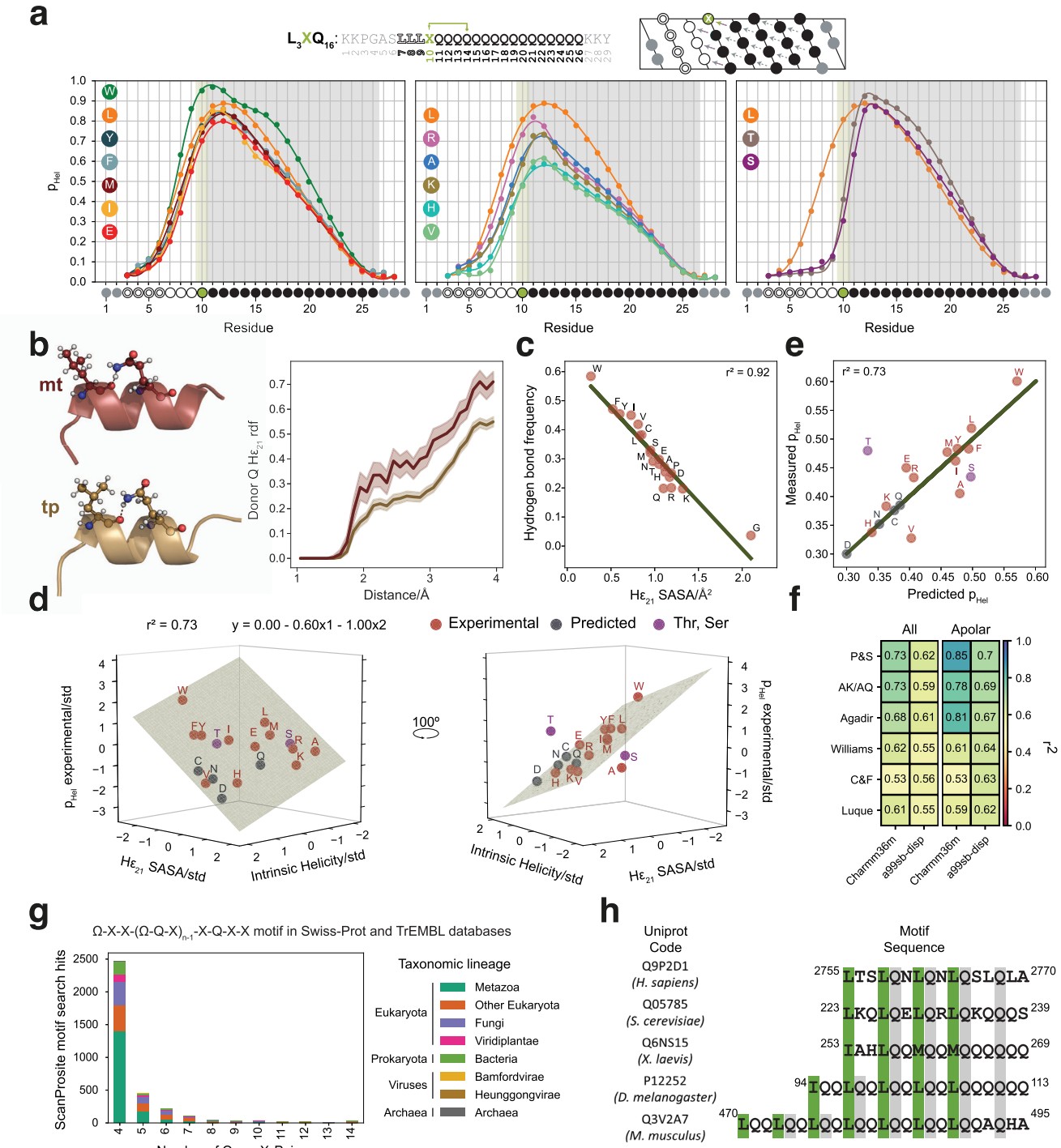

**Fig. 3 | Gln side chain to main chain hydrogen bonds can be accepted by different residues. a** Top: Sequence, numbering, and representation, as helical projection, of the $L_3XQ_{16}$ variants studied in this work. Bottom: residue-specific helical propensity of the $L_3XQ_{16}$ variants. The type of residue X (position 10) is indicated by the colored circles. Left: helical profile of the 7 most helical single variants. Center: helical profile of the least helical single variants and $L_4Q_{16}$. Right: Helical profile of the outlier variants (X = T, S) and $L_4Q_{16}$ (spectra in Supplementary Fig. 4b). **b** Effect of the rotameric state of the acceptor on the interaction of atom $H_{\varepsilon 21}$ with $H_2O$. Left: two frames of the $L_4Q_8$ Charmm36m trajectory showing Gln4 involved in a bifurcated hydrogen bond with Leu 4 in either the *mt* (top) or *tp* (bottom) rotamer. Right: radial distribution function for Gln4 $H_{\varepsilon 21}$ in the frames where Leu 4 populates the *mt* (red) or *tp* (gold) rotamer (shades show the 95% CI obtained from 10 block bootstrapping). **c** The frequency of the side chain to main chain hydrogen bond is strongly correlated with the solvent accessibility surface

area (SASA) of $H_{\varepsilon 21}$, which depends on the type of residue X. **d** Multiple regression correlating intrinsic helicity (x1) (Pace and Scholtz scale[29]) and SASA (x2) with the average helicity (y): the data were standardized to estimate the relative weight of each variable in defining the model. **e** Measured versus predicted average $L_3XQ_{16}$ helicity. **f** Squared *r* correlation scores ($r^2$) for the multiple regression shown in **e**, using different reported scales for intrinsic amino acid helicity and $H_{\varepsilon 21}$ SASA values derived from two sets of 1 µs MD trajectories independently generated with different force fields. The results are shown for all and apolar (L, I, V, F, Y, M, W, A) residues in position X. **g** Number of ScanProsite-identified protein sequences in UniprotKB (including the Swiss-Prot and TrEMBL databases) containing the $(P3-7)_n$ motif with an increasing number of $Gln_{i+4} \rightarrow \Omega_i$ pairs ($\Omega$ = W, L, Y, F, I, M; X = any amino acid). **h** Representative $(P3-7)_n$-like natural sequences with UniprotKB annotation score = 5.

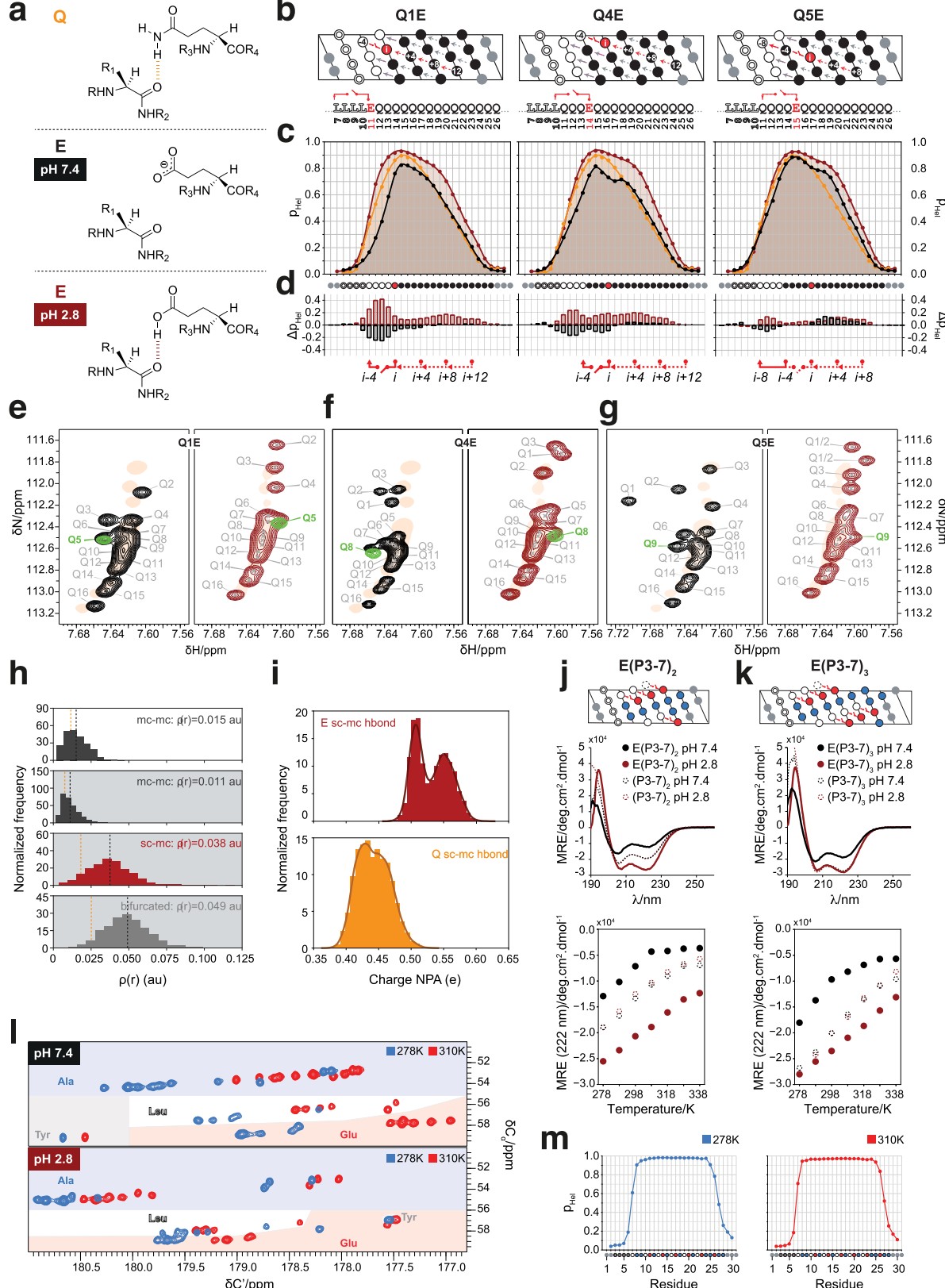

shows an alignment of some example sequences along with their UniprotKB accession code and the organism that they belong to.

### Design of a pH-sensitive conformational switch

Gln to Glu substitutions in polyQ helices decrease helical character due to the inability of the Glu side chain to donate hydrogen bonds at

physiological pH, where the carboxylate group is deprotonated[14] (Fig. 4a). Re-protonation by decreasing the pH can restore the interaction, providing us with an opportunity to introduce a pH-sensitive conformational switch in our design rules. To explore this possibility we first sought to establish whether the loss of helicity upon substitution and its restoration upon pH decrease is strictly local or

**Fig. 4 | Introduction of a pH-sensitive conformational switch. a** The side chains of Gln and Glu at pH 2.8, but not that of Glu at pH 7.4 can donate a hydrogen to the main chain CO. **b** QxE variants of $L_4Q_{16}$ with pH-sensitive $Glu_{i+4} \to X_i$ interactions shown in red. **c** Helical propensities at pH 7.4 (black) and 2.8 (dark red) compared to those of $L_4Q_{16}$ at pH 7.4 (orange). **d** Residue-specific differences in helical propensity due to substitution of Gln by Glu at pH 7.4 (black) and pH 2.8 (dark red). **e**–**g** Gln side chain $N_{\varepsilon 2}$-$H_{\varepsilon 21}$ regions of the $^1$H-$^{15}$N HSQC spectra of QxE variants at pH 7.4 (left) and 2.8 (right) with the spectrum of $L_4Q_{16}$ overlaid as an orange shade. The spectra of variants Q1E and Q4E with $^{15}$N labeling of only the Gln in position $i+4$ to the mutated residue are superimposed in green. **h** QM/MM-derived hydrogen bond electron densities for the $Glu_{i+4} \to Leu_i$ interaction. Mean values for the $Gln_{i+4} \to Leu_i$ interaction[14] are shown as an orange line. First and second panels: normalized histograms showing the distribution of the electron density $\rho(r)$ of the main chain to whether its effects instead can propagate to other parts of the sequence due to cooperativity.

main chain interaction in the absence (white background) and in the presence (gray) of the side chain to main chain hydrogen bond. Third panel: electron densities for the side chain to main chain hydrogen bond. Fourth panel: electron densities for the bifurcated hydrogen bond. **i** Natural population analysis (NPA) charges on the donor hydrogen atom in Glu and Gln, showing that its charge depends on whether it participates in the hydrogen bond (lower values) or not. **j, k** pH-dependent helicity and thermal stability for peptide $E(P3\text{-}7)_n$. Top: Helical projections. Center: CD spectra (278 K) of peptides $E(P3\text{-}7)_n$ and $(P3\text{-}7)_n$ at the indicated conditions. Bottom: thermal denaturation of peptides $E(P3\text{-}7)_n$ and $(P3\text{-}7)_n$ monitored by measuring the mean residue ellipticity at 222 nm. **l** 2D CACO NMR spectra of $E(P3\text{-}7)_3$. **m** Residue-specific helical propensities for peptide $E(P3\text{-}7)_3$ at pH 2.8.

For this, we compared the residue-specific helicity of polyQ $L_4Q_{16}$ variants with Gln to Glu substitutions at positions 1 (Q1E), 4 (Q4E), and 5 (Q5E) in the polyQ tract (Fig. 4b and Supplementary Fig. 8a). In Q1E the positions experiencing the strongest decrease in helicity are Leu 2 to Leu 4, which are tethered by the first bifurcated $Gln_{i+4} \to Leu_i$ hydrogen bond in $L_4Q_{16}$, as expected, but there is also a small decrease in helicity for Gln residues in positions 5 to 8 (Fig. 4d, left, black). Similar effects were observed in Q4E and the loss of helical character in Q5E was much smaller, likely because the interaction broken upon substitution, accepted by Gln1, is weak even at physiological pH. This indicates that breaking the first $Gln_{i+4} \to Leu_i$ interaction also weakens the interaction to which it is concatenated in the polyQ helix, and vice-versa, again in line with the notion that these two interactions form cooperatively (Fig. 4c, d, black).

Experiments at pH 2.8, where the carboxylate group of the Glu side chain is protonated, showed that helicity was even higher than that of peptide $L_4Q_{16}$ at physiological pH (Fig. 4c, d, red); in addition the dispersion of Gln side chain $N_\varepsilon$ and $H_{\varepsilon 21}$ resonances in the 2D $^1$H$^{15}$,N HSQC spectrum, that is characteristic of polyQ helices, was restored (Fig. 4e–g). To investigate the physical basis of this, we simulated this interaction by using a hybrid QM/MM approach (Supplementary Fig. 8b, c). We found that the establishment of the side chain to main chain hydrogen bond weakened the main chain to main chain hydrogen bond: the associated average electron density decreased from 0.015 a.u. to 0.011 a.u. (Fig. 4h) and the main chain $Glu_{i+4}$ (H) · $Leu_i$ (O) interatomic distance increased by 0.15 Å (Supplementary Fig. 8d). Consistent with the stronger hydrogen bond donor character of protonated Glu, relative to that of Gln, the average electron density of the $Glu_{i+4} \to Leu_i$ side chain to main chain hydrogen bond (0.038 a.u.) was higher than that involving Gln side chains (0.017 a.u.) and the electronic polarization of the $H_{\varepsilon 2}$ donor was stronger for Glu relative to Gln (Fig. 4i).

Finally, to test whether switchable side chain to main chain hydrogen bonds can be integrated in our design rules, we studied variants of the $(P3\text{-}7)_2$ and $(P3\text{-}7)_3$ peptides where all Gln residues were substituted by Glu, namely $E(P3\text{-}7)_2$ and $E(P3\text{-}7)_3$ (Fig. 4j, k, top). As expected, these were less helical than $(P3\text{-}7)_2$ and $(P3\text{-}7)_3$ at physiological pH, but more at pH 2.8 (Fig. 4j, k, center); importantly, this increase of helicity and thermostability was not observed for peptides $(P3\text{-}7)_2$ and $(P3\text{-}7)_3$, that are stabilized by Gln side chain hydrogen bonds that are not affected by pH changes. Both the switchable nature and the increased strength of the side chain to main chain hydrogen bonds involving Gln is apparent in the enhanced thermal stability of the $E(P3\text{-}7)_n$ variants (Fig. 4j, k, bottom): while at physiological pH peptide $E(P3\text{-}7)_3$ loses its helical character at a lower temperature than $(P3\text{-}7)_3$, at pH 2.8 it remains helical even at the highest temperature studied, 340 K. Indeed $^{13}$C-detected 2D CACO NMR spectra confirmed that, at pH 2.8, $E(P3\text{-}7)_3$ was highly helical at 310 K, the physiological temperature (Fig. 3l, m).

## Combining $Gln_{i+4} \to X_i$ and electrostatic interactions

The natural single α-helices studied until now are stabilized by numerous electrostatic interactions between side chains of opposite charge at relative positions $i,i+3$ or $4$[34]. We sought to investigate whether these electrostatic interactions can be combined with $Gln_{i+4} \to X_i$ side chain to main chain hydrogen bonds to stabilize α-helices. For this we studied the polyQ tract of the TATA-box binding protein (TBP), which has a primary structure that suggests the presence of an electrostatic interaction between either of two Glu residues immediately flanking the tract at the N-terminus (Glu9 and Glu10) and an Arg interrupting it (Arg13) (Fig. 5a). This interaction can occur concomitantly with two strong bifurcated hydrogen bonds accepted by Ile7 and Leu8, at position $i$-4 relative to the first two Gln residues of the tract. As observed for the polyQ tracts in AR[14] and, to a lesser extent, huntingtin[35] the CD spectrum of a monomeric (Supplementary Fig. 9) peptide spanning a tract of size 16 and its N-terminal flanking region, TBP-$Q_{16}$, showed it is strongly helical and that its expansion to 25 Gln residues increases its helicity (Fig. 5a, b).

We then used NMR to characterize TBP-$Q_{16}$ with residue resolution (Fig. 5d, e). At physiological pH, in agreement with the CD data, the peptide forms a fully folded helix between residues Glu9 and Gln14 and its helicity decreases progressively towards the C-terminus; at acidic pH, instead (Fig. 5c), at which Glu side chains are protonated, the helical propensity starts decreasing at position 12. In addition, both the spectral signature of concatenated $Gln_{i+4} \to X_i$ interactions (Fig. 5f) and the rotamer selection associated with these interactions (Fig. 5g) are diminished for Gln14 ($H_\gamma$) and, to some extent, Gln12 ($H_\beta$). These results are in agreement with the formation of a helix-stabilizing electrostatic interaction between Glu9 (or Glu10) and Arg13 that is lost upon protonation at low pH, confirming that electrostatic interactions can be combined with Gln side chain to main chain interactions. To further confirm that these interactions can co-exist we simulated a WTE-enhanced MD trajectory of TBP-$Q_{16}$[27,36]: Fig. 5h shows two frames of the trajectory in which both the $Gln11 \to Ile7$ and the $Gln12 \to Leu8$ bifurcated hydrogen bonds occur simultaneously with a salt bridge involving Arg13 and either Glu10 (left) or Glu9 (right).

## Targeting the helices to globular domains

Gln-based α-helical peptides can be modified to bind specific globular targets: the Ala residues in the $(P3\text{-}7)_n$ scaffolds (Fig. 1) can indeed be modified at will because they are not involved in the interactions that stabilize the helical structure. To prove this concept we modified the sequence of peptide $(P3\text{-}7)_3$ to interact with the C-terminal domain of RAP74 (RAP74-CTD), a small globular domain that binds to intrinsically disordered motifs that fold upon binding[37–39]. We blended the sequence of $(P3\text{-}7)_3$ with that of two different motifs (centFCP1 and cterFCP1) derived from FCP1 that interact with this globular protein independently[39]. This led to peptides δ and δ$_{ctrl}$: δ was designed to bind to RAP74-CTD whereas δ$_{ctrl}$ is a control sequence equivalent to δ where we replaced Leu by Ala that, despite having high intrinsic helical

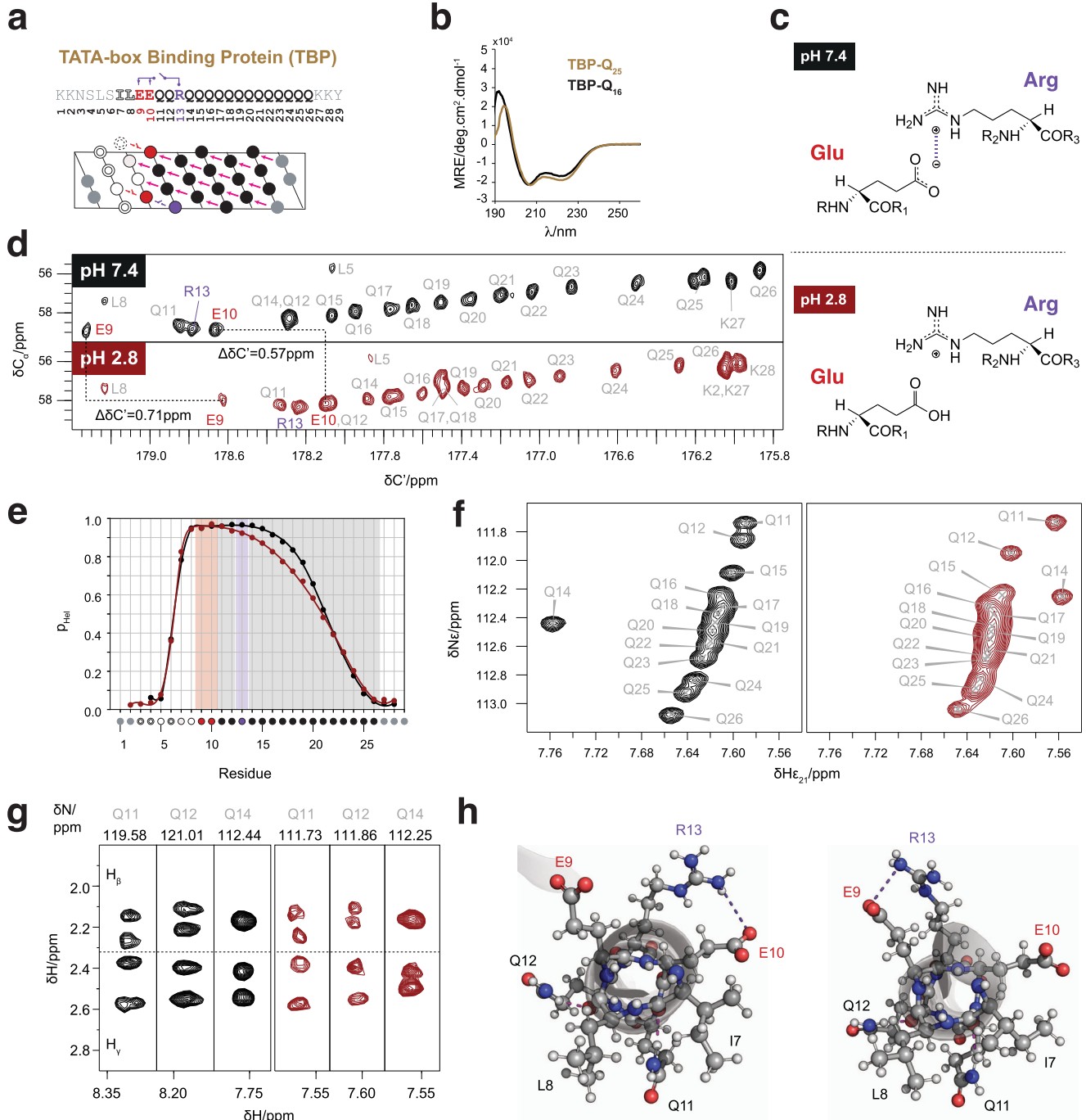

**Fig. 5 | The polyQ tract of TBP forms a helix stabilized by Gln side chain to main chain hydrogen bonds and an electrostatic interaction. a** Sequence, numbering and helical projection of peptide TBP-Q$_{16}$. The electrostatic interaction is color-coded in purple. **b** CD spectra of peptides TBP-Q$_{16}$ (black) and TBP-Q$_{25}$ (gold). **c** Scheme of the electrostatic interaction between Arg and Glu at physiological pH and its absence at acidic pH. **d** $^{13}$C-detected CACO spectra of the TBP polyQ tract at pH 7.4 (top, black) and 2.8 (bottom, dark red). **e** Residue-specific helicity of TBP at pH 7.4 (black) and 2.8 (dark red). Shades color-coded as in **a** are shown to guide the eye. **f** Region of the $^{1}$H-$^{15}$N HSQC spectra of TBP at pH 7.4 (left, black) and 2.8 (right, dark red) showing the N$_{\varepsilon2}$-H$_{\varepsilon21}$ correlations. **g** Strips from the 3D H(CC)(CO)NH spectra of TBP at pH 7.4 (left, black) and 2.8 (right, dark red) displaying the side chain aliphatic $^{1}$H resonances of the first three Q residues in the polyQ tract. Strips were chosen from either N$^{H}$ or N$_{\varepsilon2}$ for clarity. **h** Frames from the MD trajectory obtained for TBP. An orthogonal view of the helix is shown. In the left panel, a frame is shown where Arg13 (*i*) establishes an electrostatic interaction with E10 (*i-3*), represented by the purple dashed line. Simultaneously, both Gln11 and Gln12 establish bifurcated hydrogen bonds with Ile7 and Leu8, respectively (pink dashed lines). Right: a frame where Arg13 (*i*) establishes an electrostatic interaction with Glu9 (*i-4*), while the two bifurcated hydrogen bonds previously described are also present.

propensity, are bad acceptors of the side chain to main interactions, thus decreasing helicity (Fig. 6a and Supplementary Fig. 10). To facilitate the comparison with established helix-stabilization methods, we also designed δ$_{Stpl}$, which features a chemical staple covalently linking the side chains of residues at positions 13 and 17 and where Leu

hydrogen bond acceptors were substituted by Ala to weaken Gln side chain to main chain interactions.

An analysis of the structural properties of peptides δ and δ$_{ctrl}$ by CD showed, as expected, that the former is more helical than the latter, especially at room temperature (298 K), further confirming the

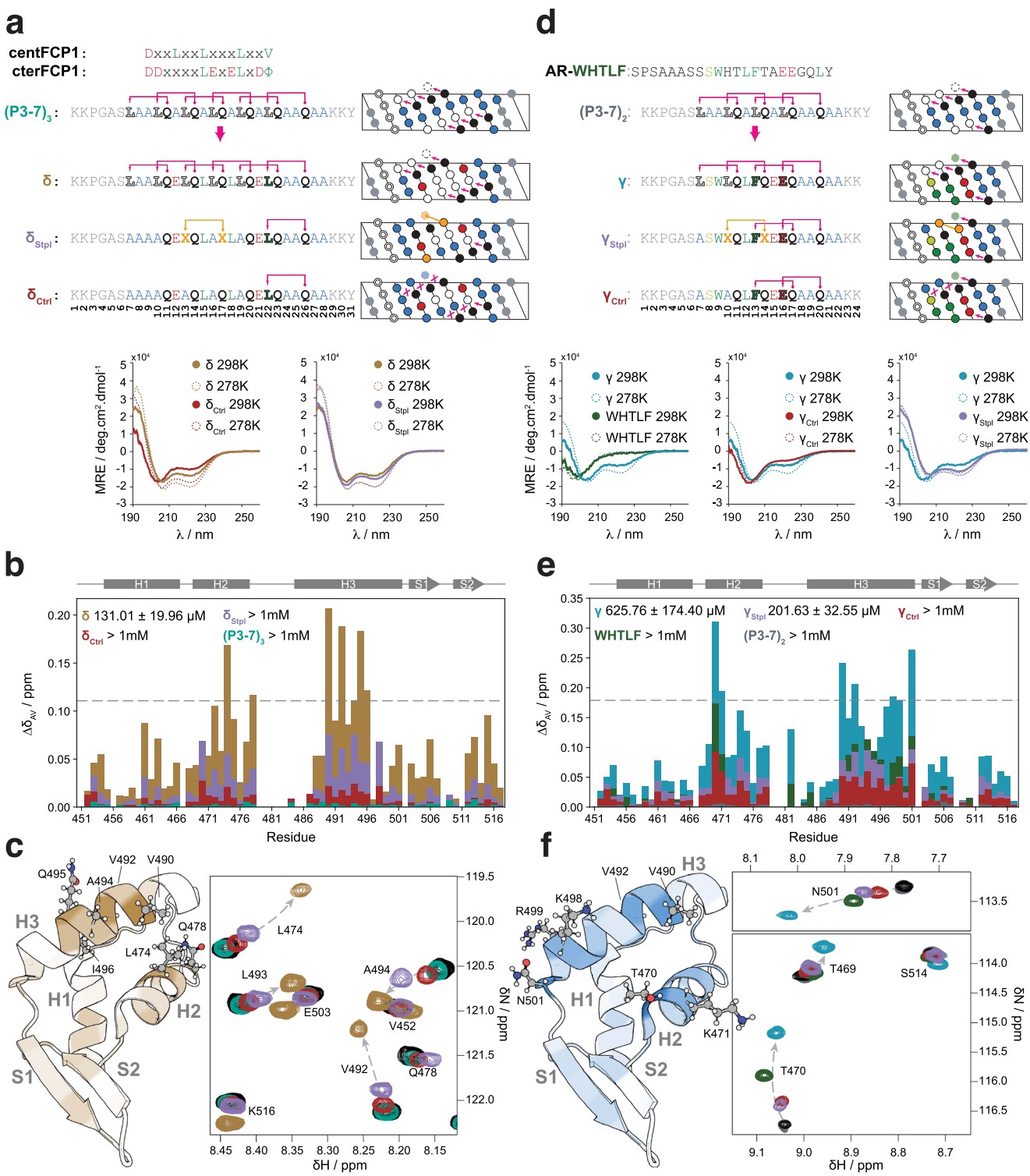

important role of Leu residues for the stability of the helical fold (Fig. 6a). At 278 K, δ and δ$_{Stpl}$ displayed equivalent helicity, although the later was slightly more helical at 298 K. We then analyzed the chemical shift perturbations (CSPs) in the $^1$H,$^{15}$N BEST-TROSY spectrum of RAP74-CTD induced by peptide binding. We obtained that δ induced perturbations in residues of the globular protein that define the binding site of FCP1 in this globular target[39], confirming a similar binding mode ($K_D = 131.01 \pm 19.96$ μM, Supplementary Fig. 11a); this interaction was also studied by isothermal titration calorimetry (ITC, Supplementary Fig. 11b). This was in contrast to the results obtained with both δ$_{ctrl}$ and (P3-7)$_3$, which in both cases failed to interact

(Figs. 6b, c). δ$_{Stpl}$, instead, induced intermediate CSPs reporting on binding in the millimolar range (Supplementary Fig. 11a). The result obtained with δ$_{ctrl}$ indicates that the helical character of δ is key for its ability to interact with RAP74-CTD, whereas those obtained with (P3-7)$_3$ indicate high helical character does not suffice, and that the identity of the residues placed in the vacant positions is indeed key for binding, in agreement with our hypothesis.

To provide a second proof of concept we blended the sequence of (P3-7)$_2$ with that of a motif found in the activation domain of AR that also interacts with RAP74-CTD, to yield peptide γ (Fig. 6d and Supplementary Fig. 10). As we previously showed, inhibiting this

**Fig. 6 | Design of Gln-based single α-helical peptides that interact with RAP74-CTD. a** Top: design of peptide δ, blending the consensus RAP74-CTD binding motifs of FCP1 with peptide (P3-7)$_3$. Peptide δ$_{Stpl}$ features a chemical staple covalently linking the side chains at positions 13 and 17, whereas all Leu residues outside the binding motif were substituted by Ala. Peptide δ$_{ctrl}$ is devoid of helix-stabilizing interactions. Bottom: CD spectra of peptides δ (gold), δ$_{Stpl}$ (purple), and δ$_{ctrl}$ (dark red) at 298 K (solid) and 278 K (dashed). **b** Averaged chemical shift perturbations observed in the $^1$H-$^{15}$N resonances of the RAP74-CTD upon the addition of 15 molar equivalents of the (P3-7)$_3$ (teal), δ (gold), δ$_{Stpl}$ (purple), and δ$_{ctrl}$ (dark red) peptides. NMR-derived K$_D$ values (Supplementary Figure 11a) are shown in the legend. The dashed line sets the threshold for the top 10% peaks with the most intense averaged CSPs upon the addition of δ. **c** Left: structure of the RAP74-CTD domain of TFIIF (PDB entry 1NHA) colored in a gradient representing the averaged CSPs observed upon the addition of 15 molar equivalents of

the δ peptide (white, minimum CSP$_{av}$; gold, maximum CSP$_{av}$). Right: region of the overlaid $^1$H-$^{15}$N BEST-TROSY spectra of RAP74-CTD in the absence (black) and in the presence of 15 molar equivalents of the (P3-7)$_3$ (teal), δ (gold), δ$_{Stpl}$ (purple), and δ$_{ctrl}$ (dark red) peptides. **d** Top: design of peptide γ blending the RAP74-CTD binding motif AR-WHTLF with peptide (P3-7)$_2$. Peptide γ$_{Stpl}$ features a chemical staple covalently linking the side chains at positions 10 and 14, whereas all Leu residues outside the binding motif were substituted by Ala. Peptide γ$_{Ctrl}$ is devoid of helix-stabilizing interactions. Bottom: CD spectra of peptides AR-WHTLF (green), γ (blue), γ$_{Stpl}$ (purple), and γ$_{Ctrl}$ (dark red) at 298 K (solid) and 278 K (dashed). **e** as in **b** for (P3-7)$_2$ (dark gray), AR-WHTLF (green), γ (blue), γ$_{Stpl}$ (purple), and γ$_{Ctrl}$ (dark red). **f** Left: as in **c**, left, for peptide γ. Right: as in **c**, right, for the (P3-7)$_2$ (purple), AR-WHTLF (green), γ (blue), γ$_{Stpl}$ (purple), and γ$_{Ctrl}$ (dark red) peptides.

interaction with small molecules or peptides is a potential avenue to treat castration-resistant prostate cancer[37]. Linear peptides spanning the AR motif bind weakly to RAP74-CTD due, at least in part, to their low helical propensity, providing us with an additional opportunity to test the potential of our designs. In peptide γ two Gln$_{i+4}$ → X$_i$ hydrogen bond acceptor positions were modified to accommodate the binding motif following the rules learned previously (Fig. 3). We also designed peptides γ$_{Stpl}$ and γ$_{Ctrl}$, analogous to their δ counterparts. CD spectra show higher helicity of γ when compared to either the WHTLF or the γ$_{Ctrl}$ peptides. In contrast, γ$_{Stpl}$ is the most helical peptide of the series. In agreement with our hypothesis, we obtained that peptide γ binds RAP74-CTD with a K$_D$ in the mid-micromolar range, at least an order of magnitude more strongly than either the (P3-7)$_2$, the WHTLF or the γ$_{Ctrl}$ peptides (Fig. 6e, f and Supplementary Fig. 11c). γ$_{Stpl}$ also binds RAP74-CTD in the mid-micromolar range (Supplementary Fig. 11c), and likely compensates its rigidity with its higher helicity when compared to γ. In summary, we have shown that the sequences of Gln-based single alpha-helices can be modified to interact with a specific globular protein.

## Discussion

Our results show that Gln side chain to main chain hydrogen bonds can be used to design linear peptides that fold into α-helices (Fig. 2) with properties that make them attractive for various applications: they are highly soluble, even upon thermal denaturation, are not stabilized by electrostatic interactions, unlike the Glu and Lys/Arg-rich single α-helices reported until now[40–42]; and display some degree of folding cooperativity due to the concatenation of side chain to main chain hydrogen bonds explicit in our design rules (Fig. 1). Our data sheds light on the potential basis of such cooperative effect, which occurs when the donor of a Gln$_{i+4}$ → Leu$_i$ interaction and the acceptor of the next one share a peptide bond.

An important feature of our design rules is their versatility: the residue accepting the hydrogen bond donated by the Gln side chain can be any residue able to shield the interaction from the competition with water, such as Trp, Leu, Phe, Tyr, Met, and Ile (Fig. 3). Remarkably, we find natural sequences fulfilling our design in different kingdoms of life. We also found structural models for 2303 of these sequences in the AlphaFold Database (AFDB)[43]: in 61.4% of the cases, a DSSP analysis[44] of the AlphaFold model shows that the (P3-7)$_n$-like motif is helical, increasing to 79.8% when only motifs devoid of helix breaking residue types (P, G) in the central part of the sequence are considered (Supplementary Fig. 7c). This conclusion holds in a subset of 42 structural models calculated without multiple sequence alignments (MSAs), showing that the helicity of this motif is encoded in AlphaFold learned structural preferences (Supplementary Fig. 7d). Thus, these motifs may represent a new class of uncharged single α-helices (SAHs) that had so far remained undetected[40–42]. In addition, in suitable cases the design can be complemented by electrostatic interactions between side chains of opposite charge and Gln residues can be mutated to Glu

to introduce a pH-responsive conformational switch that uses only natural amino acids[45] and does not involve changes in oligomerization state[46] (Figs. 4 and 5).

The key feature of our scaffold design is that it defines the identity of just a fraction of the peptide residues: the rest can be chosen or optimized for specific applications. To prove this concept, we designed two peptides to bind the globular target RAP74-CTD by using an approach analogous to previous motif-grafting attempts on folded scaffolds[47] (Fig. 6). In this specific case, naively blending these sequence features with the sequence of the designed (P3-7)$_n$ scaffold proved sufficient for successful targeting. Of note, our strategy displayed similar or even superior binding compared to chemical stapling, known to impose backbone rigidity, which highlights the importance of geometric adaptability to the target surface conferred by the transient nature of the side chain to the main chain hydrogen bonds, and gives additional room for sequence optimization. In fact, even if the affinities that we have obtained in this initial exercise (Supplementary Fig. 11) can be sufficient for certain applications[48] we anticipate that it will be possible to greatly improve them by systematically searching the sequence space available using techniques for affinity maturation based on high-throughput mutational scans[49], especially when taking advantage of the versatility of our design rules. We also showed that our scaffold peptides are highly resistant to proteolytic degradation in human plasma and readily internalized by living human cells (Supplementary Fig. 2), thus overcoming some of the most important hurdles for peptide therapeutics. Although this could be related to their amino acid composition and sequence and needs to be studied on a case-by-case basis for potential hits, our design constitutes a valuable platform from which to evolve peptides with favorable pharmacokinetic properties.

The link between polyQ tract expansion and disease onset in polyQ disorders has not been established[50]. These tracts are found in intrinsically disordered regions, and much work has been devoted to investigating whether expansion changes their structural properties. The results obtained have been inconclusive: single-molecule Förster resonance energy transfer and NMR measurements showed little influence of tract length on the conformation of the tract found in huntingtin[16,51] whereas recent studies from some of us have instead shown that the helical propensity of the tract found in the AR increases upon expansion[14,15], as does the tract found in TBP (Fig. 5b).

Our results help rationalize these observations by considering that polyQ tracts are in a polyQ helix-coil equilibrium. Its position can be influenced by the residues flanking the tract at its C-terminus[16,52] but, for a given set of solution conditions and tract length, it is mainly determined by the four residues flanking the tract at its N-terminus (Fig. 3). When these are good acceptors of Gln side chain to main chain hydrogen bonds, as in AR (LLLL), the polyQ helix is favored. Instead, when only two are good acceptors as in huntingtin (LKSF), the coil is favored[16,51,53,54]. The results obtained for TBP also fit this rationale: its flanking region (ILEE) contains two good acceptors (Ile, Leu) and the

two other residues (Glu) can establish, at physiological pH, an electrostatic interaction with the Arg residue three or four positions towards the C-terminus, favoring the polyQ helix.

The polyQ helix-coil equilibrium is sensitive to solution conditions: the entropic cost of folding results in higher stability of polyQ helices at relatively low temperatures whereas high temperatures favor the coil. This contributes in part to explain the discrepancy between the results obtained with huntingtin, where the experiments were carried out at room temperature[51], and those obtained for AR, where they were instead carried out at 278 K[14]: indeed, CD studies of the structural properties of huntingtin showed an increase in helical propensity upon tract expansion at low temperature (263 K)[35]. Finally, our observation that pairs of the concatenated side chain to main chain interactions form cooperatively contributes to explaining how expansion shifts the equilibrium to the polyQ helix state both in AR[14] and in huntingtin[51].

Our results thus suggest that polyQ tracts are in a helix-coil equilibrium that is governed by the N-terminal flanking region, by solution conditions and, due to cooperativity, by tract length. Given that interaction between low-populated helical conformations of huntingtin play a role in the early stages of its aggregation into amyloid fibrils[55,56] we propose that polyQ expansion leads to the onset of Huntington's disease at least in part by stabilizing pre-nucleation oligomeric species, where the polyQ tract is partially helical, that are on-pathway to aggregation. Our proposal is in agreement with the very recently reported effects of amino acid substitutions in the N-terminal flanking region of the polyQ tract of exon 1 of huntingtin, where increases in helical propensity led to increases in aggregation propensity both in vitro and in cells[57].

In summary, we have shown how an appropriate concatenation of Gln side chain to main chain hydrogen bonds makes it possible, on the one hand, to design highly helical peptides that can be tailored to specific applications and, on the other hand, to rationalize the until now perplexing observations regarding the structural properties of polyQ tracts. We anticipate that the knowledge gained about this interaction will influence future developments in peptide design, particularly in the use of peptides as therapeutics, as well as contribute to better understanding the molecular basis of polyQ diseases.

## Methods

### Peptide sample preparation
Recombinant peptides with $^{15}N$ or $^{13}C,^{15}N$ isotope enrichment were prepared as detailed elsewhere[58]. Synthetic genes coding for the peptide of interest and codon-optimized for expression in *Escherichia coli*, with an N-terminal His$_6$-Sumo tag fusion and cloned into the pDEST-17 vector, were directly obtained from GeneArt (Thermo Fisher Scientific, Waltham, MA, USA). Genes coding for the $L_4Q_{16}$ variants in the $L_3X$ and QXE series (see Supplementary Table 2) were obtained using the Q5 Site-Directed Mutagenesis Kit from New England Biolabs (Ipswich, MA, USA). Rosetta (DE3)pLysS competent cells (Novagen, Merck KGaA, Darmstadt, Germany) were used for expression in M9 medium containing $^{15}NH_4Cl$ and, where required,$^{13}C$-glucose (both from Cambridge Isotope Laboratories Inc., Tewksbury, MA, USA) as the sole nitrogen and carbon sources, respectively. All purification steps were performed at 277 K. Cell lysates in lysis buffer (20 mM Tris-HCl, 100 mM NaCl, 20 mM imidazole, pH 8.0) were purified by immobilized metal affinity chromatography (IMAC) using a HisTrap HP 5 mL column mounted on an Äkta Purifier System (GE Healthcare, Chicago, IL, USA). Fractions containing the His$_6$-Sumo-peptide fusion in elution buffer (20 mM Tris-HCl, 100 mM NaCl, 500 mM imidazole, pH 8.0) were pooled and dialyzed overnight in lysis buffer while treated with ubiquitin-like specific protease Ulp1 (50 μg/mL). The His$_6$-Sumo tag was removed with an additional IMAC step and the peptide-containing flow-through was dialyzed in ultrapure MilliQ water before lyophilization. Unlabeled synthetic peptides from solid-phase peptide synthesis were directly obtained from Genscript (Piscataway, NJ, USA) as lyophilized powder with >95% purity. Stapled peptides $\gamma_{Stpl}$ and $\delta_{Stpl}$ were also synthesized by Genscript through the introduction at two positions, denoted as X, of an unnatural amino acid with a di-substitution at the $C_\alpha$ position (pentenylalanine). The $i, i + 4$ stapled peptides were produced by crosslinking the unnatural amino acids via ring-closing metathesis. Both recombinant and synthetic lyophilized peptides were dissolved in 6 M guanidine thiocyanate (Merck KGaA, Darmstadt, Germany) and incubated overnight at 1250 rpm, 298 K, in a thermoblock. The sample was then injected in an Äkta Purifier System equipped with a Superdex Peptide 10/300 GL column (GE Healthcare, Chicago, IL, USA) equilibrated with ultrapure water, 0.1% trifluoroacetic acid. The fractions containing monomeric peptide were pooled and centrifuged at $386,000 \times g$ for 3 h in an Optima TLX ultracentrifuge equipped with a TLA 120.1 rotor (Beckman Coulter, Atlanta, GA, USA). Orthophosphoric acid or sodium phosphate was added to a final concentration of 20 mM to adjust the pH to 2.8 or 7.4, respectively. The peptide concentration was determined by measuring the absorbance at 280 nm (extinction coefficients were calculated using the Protparam tool on the ExPASy website, https://web.expasy.org/protparam) before diluting the sample to the final experimental concentration.

### Protein sample preparation
Recombinant samples with $^{15}N$ isotope enrichment of the C-terminal domain of subunit 1 of the general transcription regulator TFIIF (RAP74-CTD), spanning residues 450-517, were obtained as described previously[37]. A codon-optimized synthetic gene cloned in a pDONR221 vector was obtained from GeneArt (Thermo Fisher Scientific, Waltham, MA, USA) and subcloned into a pDEST-His$_6$MBP vector obtained from Addgene. Rosetta (DE3)pLysS competent cells (Novagen, Merck KGaA, Darmstadt, Germany) were grown in MOPS medium with $^{15}NH_4Cl$ as the only nitrogen source (310 K, induction at $OD_{600} = 0.7$, 1 mM IPTG, harvesting 3 h after induction). Soluble fractions of cell lysates from sonication in lysis buffer (50 mM Tris-HCl, 1 M NaCl, 10 mM imidazole, pH 8.0) were purified by IMAC, and fractions containing the His$_6$MBP-TEV-RAP74-CTD fusion were dialyzed at 277 K overnight against cleavage buffer (50 mM Tris-HCl, 200 mM NaCl, 0.5 mM EDTA, pH 8.0) in the presence of TEV protease (50 μg/mL). A second IMAC step was performed and the flow-through containing the RAP-CTD was loaded onto a HiTrap SP HP cation exchange column followed by a size exclusion chromatography step on a Superdex 75 GL 10/300 column equilibrated with NMR buffer (20 mM sodium phosphate, 0.1% trifluoroacetic acid, pH 7.4), both mounted in an Äkta Purifier System (GE Healthcare, Chicago, IL, USA). The sample was concentrated to 100 μM using an Amicon Ultra 15 mL centrifugal filter (Merck KGaA, Darmstadt, Germany).

### CD spectroscopy
Peptide samples for CD spectroscopy were diluted to a final concentration of 30 μM in a volume of 400 μL in either 20 mM phosphoric acid (pH 2.8) or sodium phosphate (pH 7.4) buffer. Spectra were obtained at 278 K (unless stated otherwise) in a Jasco 815 UV spectrophotopolarimeter with a 1 mm optical path cuvette using a data interval of 0.2 nm in the 190–260 nm range with a scanning speed of 50 nm min$^{-1}$ and 20 accumulations. A blank spectrum acquired on the pertaining buffer under the same experimental conditions was subtracted from the sample spectrum. Thermal denaturation experiments were performed by acquiring a single accumulation spectrum with the same parameters at 10 K intervals, with a temperature ramp speed of 10 K min$^{-1}$ and an equilibration time of 1 min.

### NMR spectroscopy
Peptide samples for NMR spectroscopy were diluted to a final concentration of 100 μM in a volume of 400 μL in either 20 mM

phosphoric acid (pH 2.8) or sodium phosphate (pH 7.4) buffer with added 10% v/v D$_2$O and 10 μM DSS for internal chemical shift referencing, then filled into Shigemi tubes (Shigemi Co. Ltd, Tokyo, Japan). All NMR experiments were recorded at 278 K (unless stated otherwise) on either a Bruker Avance III 600 MHz or a Bruker Avance NEO 800 MHz spectrometer, both equipped with TCI cryoprobes, using TopSpin 4.0.8 for data acquisition (Bruker, Billerica, MA). Unlabeled synthetic peptides P1-5, P2-6, P3-7, P5-9 were characterized by two-dimensional homonuclear (TOCSY and NOESY) and heteronuclear ($^1$H-$^{13}$C HSQC) experiments. The TOCSY and NOESY mixing times were set to 70 and 200 ms, respectively. Water suppression was achieved by excitation sculpting[59] using a 2 ms long Squa100.1000 selective pulse. For peptide backbone resonance assignment, using uniformly $^{15}$N,$^{13}$C labeled peptides, the following series of 3D triple resonance BEST-TROSY[60] experiments were acquired with 25% non-uniform sampling (NUS): HNCO, HN(CA)CO, HN(CO)CA, HNCA, and HN(CO)CACB. For some peptides, we resolved assignment ambiguities by also acquiring a 3D (H)N(CA)NH spectrum. Furthermore, 2D $^{13}$C-detected CACO and CON experiments[61] were measured. Data processing was carried out with qMDD[62] for non-uniform sampled data and with NMRPipe[63] for all uniformly collected experiments. Data analysis was performed with CcpNmr Analysis version 2.4[64]. Determination of the residue-specific helical propensity ($p_{Hel}$) from backbone chemical shifts H$^N$, N$^H$, C', and C$_\alpha$ was performed using CheSPI[65], which uses sequence and condition-corrected (temperature, pH) estimates for the reference random coil chemical shifts derived from POTENCI[66]. CheSPI was chosen over other algorithms as it intrinsically considers chemical shift changes derived from Glu side chain protonation, as the -0.6 ppm C' chemical shift change reported before[67] and observed in the TBP CACO spectra (Fig. 5d).

Side chain aliphatic $^1$H chemical shifts were obtained from 3D $^{15}$N-edited TOCSY-HSQC (75 ms mixing time) and NOESY-HSQC (200 ms mixing time) spectra. Glutamine side chain resonances were assigned using complementary 3D H(CC)(CO)NH and (H)CC(CO)NH spectra recorded with 25% NUS and 14 ms C,C-TOCSY mixing. To further confirm the side chain N$_\varepsilon$ assignments for Gln5 in peptide Q1E and Gln8 in peptide Q4E we recorded 2D $^1$H-$^{15}$N HSQC spectra of synthetic unlabeled peptides with specific $^{15}$N labeling only in the positions of interest.

For the detailed analysis of side chain rotamer distributions by the CoMAND approach, a 3D CNH-NOESY (i.e., 3D [H]C,NH HSQC-NOESY-HSQC) spectrum[68] of [U-$^{13}$C$^{15}$,N] labeled (P3-7)$_2$ was recorded at 800 MHz, 278 K, with 400 ms NOE mixing time and 64($^{15}$N) × 86($^{13}$C) × 2048($^1$H) complex data points corresponding to 15.2 × 104.8 × 6.3 Hz FID resolution. The final $^{15}$N-HSQC module employed sensitivity-enhanced coherence selection by gradients and band-selective flip-back of H$^C$ polarization to enable its faster re-equilibration during a shorter total interscan delay (experimentally optimized as 0.6 s). For the prior assignment of all aliphatic side chain $^1$H and $^{13}$C resonances and easy distinction between intra- and inter-residual NOE signals in the [H]C,NH HSQC-NOESY-HSQC spectrum, we furthermore recorded a set of 3D [H]CC[CA]NH TOCSY (11.3 and 22.6 ms FLOPSY8 mixing) and [H]CC[CO]NH TOCSY spectra (9 and 18 ms FLOPSY8 mixing).

To study glutamine side chain $^{15}$N relaxation, a $^{15}$N labeled (P3-7)$_2$ sample was prepared in NMR buffer (pH 7.4). To avoid bias due to dipole-dipole cross-correlated relaxation within the N$_{\varepsilon2}$H$_2$ moieties and thus allow a direct comparison with the main chain NH data, we sampled only their 50% semi-protonated N$_{\varepsilon2}$HD isotopomers in buffered 50% D$_2$O and applied continuous deuterium decoupling during the $^{15}$N coherence evolution. Of note, the differential deuterium isotope shift of $^{15}$N$_{\varepsilon2}$ in the N$_{\varepsilon2}$H$_{\varepsilon21}$D$_{\varepsilon22}$ vs N$_{\varepsilon2}$D$_{\varepsilon21}$H$_{\varepsilon22}$ species[69] also allowed an unambiguous stereospecific signal assignment of the attached side chain carboxamide H$_\varepsilon$ (Supplementary Fig. 3b). To measure $^{15}$N R$_1$ and R$_2$ rates the conventional pulse sequences with sensitivity-enhanced coherence selection by gradients, water flip-back, and fully interleaved

acquisition of relaxation delays were complemented with continuous deuterium decoupling during t$_1$($^{15}$N) in order to suppress $^{15}$N(t$_1$) line broadening from scalar relaxation (via $^1$J$_{15N,D}$ coupling) for the glutamine side chain NHD isotopomers of interest. An exponential decay function was fitted to the data to obtain T$_1$ and T$_2$ values:

$$I_t = I_0 e^{\left(\frac{-t}{T_{1,2}}\right)} \quad (1)$$

with $I_O$ and $I_t$ corresponding to peak intensity at times 0 and $t$, respectively. R$_2$/R$_1$ ratios were calculated as T$_1$/T$_2$, and errors were derived by propagating the SD of the fits. In contrast, the pulse sequence for measuring the $^{15}$N{$^1$H} heteronuclear NOE (likewise with sensitivity-enhanced coherence selection by gradients and continuous deuterium decoupling during t$_1$($^{15}$N)) required further critical adaptations to suppress detrimental antiphase signal components (in F1($^{15}$N)) for the 50% glutamine NH$_2$ isotopomers that impede a clean quantification of nearby NHD signals of interest. Thus, for the reference (non-saturated) spectrum, the first 90° $^1$H pulse in the sensitivity-enhanced reINEPT following t$_1$($^{15}$N) had to be cycled (inverted) along with the receiver phase. For the H$^N$ saturated spectrum, however, further antiphase contamination derives from some 4N$_Z$H$^+$H$'^±$ multi-quantum coherence forming during the H$^N$ saturation sequence[70] that can be removed by its phase cycling and/or by appending a concatenated $^1$H spoil sequence (z-gradient 1 − 90°($^1$H) - z-gradient 2). NOE SD values were calculated as previously described[71].

RDCs were determined from the difference between couplings observed for aligned versus unaligned (P3-7)$_2$ samples (with U-$^{15}$N,$^{13}$C labeling) where alignment was achieved[72] using a gel kit from New Era Enterprises, Inc. (Vineland, NJ, USA). For this, 7% acrylamide gels were dialyzed in ultrapure water for 3 h and NMR buffer overnight. The prepared gels were then soaked in the peptide sample (ca. 0.2 mM) overnight at 277 K and squeezed into open-ended 5 mm NMR tubes using a funnel and piston. The filled tube was closed with a bottom plug and a Shigemi top plunge. The sample alignment uniformity was assessed via the deuterium signal splitting. One bond $^1$H-$^{15}$N RDCs were obtained from comparing aligned versus unaligned 2D BEST-TROSY spectra[60] measured at 278 K and selecting either the H$^N$ TROSY or semi-TROSY signals in the direct dimension. One bond $^{13}$C'-$^{15}$N and two bond $^{13}$C'-$^1$H$^N$ RDCs were derived by comparing the pertaining $^1$J$_{C',N}$ splitting in the indirect ($^{15}$N) and $^2$J$_{C',H}$ splitting in the direct (H$^N$) dimension, respectively, observed in the (not $^{13}$C decoupled) 2D $^1$H,$^{15}$N HSQC spectra of aligned vs unaligned samples measured at 278 K. PALES[73] was used to calculate the expected RDCs for each structure in the CoMAND-derived ensemble, allowing for the calculation of an independent alignment tensor for each frame based only on its coordinates. Predicted RDCs were obtained as the average of the ca. 200 frames generated in the 20 iterations of converged, R-factor-minimizing CoMAND global ensemble calculations, and scaled to minimize the RMSD against experimentally determined RDCs.

To study the interaction between the peptides and RAP74-CTD we measured the 2D $^1$H-$^{15}$N BEST-TROSY spectrum of $^{15}$N-labeled RAP74-CTD (50 μM throughout, 298 K) in the presence of increasing concentrations of peptide. Before the experiment, both the protein and the peptide were dialyzed (277 K, two dialysis steps) in the same preparation of NMR buffer (20 mM sodium phosphate, 0.1% trifluoroacetic acid, pH 7.4), using a Pur-A-Lyzer (Sigma-Aldrich, Burlington, MA, USA) and a Micro Float-A-Lyzer (Spectrum Laboratories, San Francisco, CA, USA) respectively. The chemical shift assignment of RAP74-CTD was reported previously (BMRB code 27288). Averaged $^1$H and $^{15}$N chemical shift perturbations (CSPs) were calculated as:

$$\Delta\delta_{AV} = \sqrt{(\Delta\delta_{1H})^2 + \left(\frac{1}{5}\Delta\delta_{15N}\right)^2} \quad (2)$$

Dissociation constants, $K_D$, and averaged CSP amplitude, $\Delta\delta_{max}$, for $\delta$, $\delta_{stpl}$, $\gamma$ and $\gamma_{stpl}$ peptide binding to RAP74-CTD were obtained by a global fitting (nonlinear regression) of the 10% peaks with the largest averaged CSPs to the following single-site binding model (1:1 stoichiometry):

$$\Delta\delta = \Delta\delta_{max} \frac{([Protein] + [Peptide] + K_D) - \sqrt{([Protein] + [Peptide] + K_D)^2 - 4[Protein][Peptide]}}{2[Protein]}$$

(3)

A Monte-Carlo simulation varying both protein and peptide concentrations within 20% experimental errors was used to derive error margins for the final $K_D$ values.

### Structural characterization using CoMAND

To investigate the conformational tendencies of the $(P3\text{-}7)_2$ peptide we applied the CoMAND method (Conformational Mapping by Analytical NOESY Decomposition[24]). This method analyzes a 3D CNH-NOESY spectrum (i.e. 3D [H]C,NH HSQC-NOESY-HSQC), which displays only NOE contacts between $^{15}$N-bound and $^{13}$C-bound protons and is therefore intrinsically diagonal-free[68]. As a first step, one-dimensional $^{13}$C sub-spectra (strips) were extracted from this spectrum ($\tau_m$=400 ms). Each strip is taken perpendicular to a specific $^{15}$N-HSQC position and represents contacts to a single $^{15}$N-bound proton, edited by the $^{13}$C shift of the attached carbon. For $(P3\text{-}7)_2$ we obtained strips for 18 main chain amide protons (residues L7 to K24) and all 4 glutamine side chain $H_{\varepsilon21}$ protons (Q11, Q14, Q17, Q20). These strips are analyzed in terms of a quantitative R-factor expressing the agreement between experimental and back-calculated spectra. Global back-calculation parameters for CoMAND were optimized by grid searching, resulting in an overall correlation time of 2.0 ns and effective $^{13}$C signal halfwidth of 14 Hz.

For a reconstruction of the experimental $^{13}$C strips, CoMAND compiles a linear combination of strips back-calculated from a set of trial conformers that should reflect the conformational space of each residue. Here, we used the a99sb-disp and DES-amber MD trajectories and back-calculated 13002 frames from each trajectory for each of the 22 experimental strips. For each residue, the conformational ensemble producing the lowest R-factor was then compiled using the CoMAND stochastic optimization method[24]. A starting conformer is randomly selected from the 20 conformers with lowest R-factors. All conformers are then tested in random order, with a new member added to the ensemble if it decreases the R-factor by more than a given threshold (0.0005). Convergence is achieved if no further conformer is found or if the ensemble reaches a maximum size, here set at 20 structures. Due to its stochastic nature, this selection procedure can be repeated to produce ensembles with similar R-factors, but sampling a wider range of conformers.

For $(P3\text{-}7)_2$ we applied a two-step protocol for each residue. In the first step we established the minimum R-factor by compiling 100 per-residue ensembles, optimizing over single experimental strips. These per-residue ensembles were also used to define a set of *witness* strips; i.e. those whose R-factors may be affected by conformational changes in the residue in question. In the second step, we obtained the conformational distribution by co-optimizing over these witness strips. For each set, 100 optimization trials were run for each MD trajectory frame pool, resulting in 100 ensembles per frame pool, each typically containing 5–15 members. After removing ensembles with R-factors significantly above average (90% confidence interval), the set of conformers used in co-optimization (typically over 1000) was pooled to represent the conformational diversity for each residue.

To quantify the conformational distributions, we clustered the data via GMM. A vector of $n$ features - here dihedral angles - was defined for each conformer which was then used to train a model describing the probability $p(k|x)$ that a data point $x$ is a member of

cluster k. For each cluster, this probability is defined by an $n$-dimensional multivariate Gaussian distribution representing its center and shape, and by a prior probability, $p(k)$, corresponding to its relative population. These model parameters were fitted to the training data using the Expectation Maximization (EM) algorithm, modified to accommodate the periodic nature of dihedral angles. For $(P3\text{-}7)_2$, we applied the GMM method for the $\chi_1/\chi_2$ pairs of all leucine and glutamine residues. As the number of clusters that best describe the training data was not known a priori, we searched values from 1 to 9 systematically and assessed the fit via the Bayesian information criterion, a measure that includes a penalty for model complexity. EM initialization requires an arbitrary seed value for each cluster center. For $\chi_1/\chi_2$ pairs, it is convenient to select seeds at the center of a rotameric form, with seeds progressively added with increasing cluster number, according to their database frequency. The best scoring model was stored for each residue.

The GMM method provides a compact but detailed description of conformational landscapes for use in downstream calculations. Here we have applied it to Monte-Carlo conformational sampling as part of an extended "greedy" R-factor optimization protocol. The pooled a99sb-disp and DES-amber MD trajectories were systematically sampled and the conformer affording the greatest reduction in global R-factor was added in each iteration. Thus, 2–4 conformers were typically added to the ensemble, which was then further optimized by adjusting the side chain conformations for leucine and glutamine residues using $\chi_1/\chi_2$ combinations from the corresponding GMM model with a 0.05 probability cutoff. For each residue in the ensemble, up to 30 $\chi_1/\chi_2$ combinations were tested to find sterically acceptable conformations lowering the global R-factor. For glutamine, the $\chi_3$ angle was additionally sampled around population centers pertaining to each $\chi_1/\chi_2$ combination (five trials; standard deviation 8°). Note that enthalpic contributions from hydrogen bonding were not considered in testing conformers, such that their selection was primarily driven by the reduction in R-factors. This iterative process of ensemble selection and modification was repeated until no further conformers were added by the greedy step. This protocol was repeated 20 times to probe the consistency of results and an example was chosen as the final ensemble (Fig. 2g).

### Molecular dynamics simulations

The trajectories for the $(P3\text{-}7)_2$ peptide used in the CoMAND analysis and for the TBP peptide shown in Fig. 5h were generated with the Well-Tempered Ensemble (WTE)[36,74] enhanced sampling algorithm starting from a fully helical conformation. We used 26 energy-biased replicas within the temperature range from 275 K to 500 K, and two unbiased replicas at 278 K and 298 K. The unbiased replica at 278 K (at which temperature the 3D [H]C,NH HSQC-NOESY-HSQC was measured) was then used to generate the conformations for our CoMAND analysis and Fig. 5h. For the biased replicas, the energy bias was increased during the first 500 ps and then kept constant. During the bias-deposition, a Gaussian with a height of 1.2 kJ mol$^{-1}$ and a width of 140 kJ mol$^{-1}$ was added every 0.5 ps. The bias factor was set to 16. All replicas were subsequently used in a production simulation for 200 ns, where conformations used by CoMAND were extracted from the last 150 ns. The exchange between replicas was monitored to ensure good replica diffusion in temperature space. The production simulation was run in the NPT ensemble. In order to generate frame pools for our CoMAND analysis that best reproduce the experimental results, we used two recent force fields for these simulations, DES-amber[26] and a99sb-disp[27], each with its pertaining TIP4P-D water model. These are two of the best force fields describing the helix-coil equilibrium, and we used both of them to test our model for robustness. Bonds with hydrogen atoms were constrained and a time-step of 2 fs was used. For our simulations, we used Gromacs 2019.4[75-77] patched with the PLUMED library[78] version 2.5.3[79] to enable the WTE sampling method.

Two sets of trajectories for segments of $L_3X$ peptides (see Supplementary Table 2) with the sequence $L^1L^2L^3XQ^1Q^2Q^3Q^4Q^5Q^6Q^7Q^8$ (where X is any of the 20 natural proteinogenic amino acids) were calculated using either the Charmm36m force field[31] with a TIP3P[80] water model or the a99sb-disp force field[27] with its TIP4P-D water model. In this case, our objective was to obtain accurate estimates of the donor $Q_{i+4}$ $H_{\varepsilon 21}$ SASA depending on the nature and rotamer populations of the acceptor residue $X_i$. For this reason, we prioritized the choice of two force fields of different origins over an accurate description of the helix-coil equilibrium. Instead, to bias the simulations towards relevant conformations, calculations were started from a fully helical conformation, and the backbone $\varphi$ and $\psi$ dihedral angles of residues $L^2$ to $Q^5$ were restrained to $-60°$ and $-40°$, respectively, with a spring constant $k$ value optimized at $5\ kJ\ mol^{-1}\ degree^{-2}$ (Supplementary Figs. 5a, 6a) that was maintained across the trajectory. Similarly, a bias to optimize the occurrence of $Q^4 \rightarrow X$ side chain to main chain hydrogen bonds was introduced by restraining the distance between the main chain O of residue X and the $N_{\varepsilon 2}$ of residue $Q^4$ to 4 Å, with a spring constant $k$ value optimized at $50\ kJ\ mol^{-1}\ nm^{-2}$ for the Charmm36m simulations (Supplementary Fig. 5b) and $250\ kJ\ mol^{-1}\ nm^{-2}$ for the a99sb simulations (Supplementary Fig. 6b). Structure minimization and thermalization at 278 K was performed in the NVT ensemble for 1 ns. For the 1 μs NPT production runs we used Gromacs 2019.4[75–77] and an equilibration period of 100 ns that was excluded from trajectory analysis.

A trajectory was calculated for peptide Q1E using the CHARMM22*[81] force field and TIP3P water model[80] starting from a fully helical conformation to obtain different structures with $Glu_{i+4} \rightarrow Leu_i$ side chain to main chain hydrogen bonds involving the glutamic acid carboxyl group that served as seeds for the QM/MM simulations (see below). The dihedral angle $\chi_4$ orienting the glutamic acid carboxyl group was restrained to $0°$ since its most stable conformation in CHARMM22*, corresponding to $\chi_4 = 180°$, is incompatible with the experimentally indicated side chain to main chain interaction. We previously reported an equivalent QM/MM study on the $Gln_{i+4} \rightarrow Leu_i$ interaction where we used the CHARMM22* force field to generate the seeds and treat the classical sub-system[14], so an equivalent configuration allowed us the direct comparison of the hydrogen bond electron densities and NPAs shown in Fig. 4h, i. A fully helical starting structure was thermalized (300 K) and equilibrated in the NVT ensemble for 1 ns. The 1 μs production run was obtained using ACEMD[82] with a 100 ns equilibration period.

## QM/MM simulations

Four starting structures were selected from the classical MD trajectory of peptide Q1E conserving their box of water and ions. For the QM/MM simulations, we used AMBER 20[83] coupled to the QM Terachem 1.9 interface[84-86]. The QM subsystem was described at the BLYP/6-31 G* level of theory including dispersion corrections[87] and comprised 66 atoms including linker atoms. The classical subsystem was treated with the CHARMM22*[81] and TIP3P[80] force fields. The linker atom procedure was employed to saturate the valency of the frontier atoms and electrostatic embedding was used as implemented in AMBER. An electrostatic cutoff of 12 Å and periodic boundary conditions were employed throughout all QM/MM-MD simulations, using a time step of 1 fs. Structures were minimized, thermalized, and equilibrated for 10 ps at the QM/MM level prior to the 150 ps-long production runs. Finally, for each of the 150 ps QM/MM-MD runs, a Natural Bond Critical Point analysis[88,89] was performed using NBO 7.0[90].

## Database motif searches

UniprotKB[32] (including both the Swiss-Prot and the TrEMBL databases) was queried for protein sequences containing motifs that fulfill our design rules using the ScanProsite[33] motif search tool hosted at the Expasy website (https://prosite.expasy.org/scanprosite/). The query motif was introduced in Prosite format as: [LFYWIM]-X-X-[LFYWIM]-Q-X-[LFYWIM]-Q-X-[LFYWIM]-Q-X-Q-X-X-Q-X-X to quest for sequences with n = 4 $Q_{i+4}$ - [LFYWIM]$_i$ pairs, with the number of central -[LFYWIM]-Q-X- triplets increased stepwise for concomitant increases in the quested number of $Q_{i+4}$ - [LFYWIM]$_i$ pairs. UniprotKB annotation scores and taxonomic lineage information were obtained by programmatically accessing this information in UniprotKB using the accession codes from the ScanProsite searches.

## Data analysis and plotting

Data were analyzed using Python 3.7.8 along with packages Pandas 1.3.5, Numpy 1.21.2, Scipy 1.7.3, Biopython 1.76, and MDtraj 1.9.3, whereas Matplotlib 3.5.2 and Seaborn 0.11.2 were used for data plotting and visualization. PyMOL 2.3.5 was used to generate the figures displaying macromolecular structures.

## Hydrogen bond criteria

We considered two atoms to be hydrogen bonded if the distance between donor H and the acceptor O was shorter than 2.4 Å and their angle was larger than $120°$.

## Reporting summary

Further information on research design is available in the Nature Portfolio Reporting Summary linked to this article.

## Data availability

The chemical shift assignments for peptides P1-5, P2-6, P3-7, P5-9, (P3-7)$_2$, (P3-7)$_3$, u(P3-7)$_2$, u(P3-7)$_3$, $L_4Q_{16}$ (pH 2.8), $L_3WQ_{16}$, $L_3YQ_{16}$, $L_3FQ_{16}$, $L_3MQ_{16}$, $L_3IQ_{16}$, $L_3EQ_{16}$, $L_3RQ_{16}$, $L_3AQ_{16}$, $L_3KQ_{16}$, $L_3HQ_{16}$, $L_3VQ_{16}$, $L_3TQ_{16}$, $L_3SQ_{16}$, Q1E, Q4E, Q5E, E(P3-7)$_3$ and TBP in all measurement conditions specified in this work have been deposited in the BMRB (www.bmrb.io) with accession codes 51592, 51593, 51594, 51595, 51591, 51597, 51608, 51609, 51616, 51578, 51579, 51569, 51573, 51571, 51568, 51574, 51567, 51572, 51570, 51577, 51576, 51575, 51580, 51581, 51582, 51596 and 51583, respectively (Supplementary Table 2). The sequences of all oligonucleotides used in this study are available in Supplementary Table 3. Chemical shift assignments for $L_4Q_{16}$ at pH 7.4 are available at BMRB entry 27716. For peptide (P3-7)$_2$, chemical shift assignments along with relaxation data ($^{15}N$ T1,$^{15}N$ T2,$^{1}H$-$^{15}N$ heteronuclear NOEs), residual dipolar couplings (DNH, DNC, DCH, Supplementary Table 4), and spectral density peaks (J(0), J($\omega_N$)) have been deposited in the BMRB with accession code 51591. The structural ensemble of peptide (P3-7)$_2$ shown in Fig. 2g is available as Supplementary Data 1 and as PDB entry 8B1X. All 20 calculated ensembles of peptide (P3-7)$_2$ are contained in Supplementary Data 2. The solution structure of RAP74-CTD is available as PDB entry 1NHA. The molecular dynamics simulations generated in this study have been deposited in Zenodo (https://doi.org/10.5281/zenodo.7270329). The UniProt accession codes, annotation scores and taxonomic information related to proteins containing P3-7$_n$-like motifs are available in .xlsx format as Supplementary Data 3. The same file contains the accession codes related to those proteins for which we found structural models in the AlphaFold Database along with the results of the DSSP analysis. Also in the same file information is available related to the subset of proteins for which we calculated structural models using ColabFold with and without MSAs along with the results of the DSSP analysis. All structural models calculated with ColabFold are available as supplementary material as .pdb files contained in Supplementary Data 4. Source data are provided with this paper.

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

## Acknowledgements

We thank Luis Serrano for help with the Agadir predictions and helpful discussions, Ben Lehner and Ernest Giralt for helpful discussions and the ICTS NMR facility, managed by the scientific and technological centers of the University of Barcelona (CCiT UB), for their help in NMR. B.M. acknowledges funding from the Asociación Española contra el Cáncer (FCAECC project #POSTD211371MATE). C.G. acknowledges a graduate fellowship from MINECO (PRE2018-084684). M.S.-N. acknowledges funding from MINECO (PID2020-119810RB-I00). M.S.-N. holds a Ramón y Cajal contract (RYC2018-024759-I) from the Spanish Ministry of Science, Innovation, and Universities. X.S. acknowledges funding from AGAUR (2017 SGR 324), MINECO (BIO2015-70092-R and PID2019-110198RB-I00), and the European Research Council (CONCERT, contract number 648201). B.B.K acknowledges funding from the Novo Nordisk Foundation (#NNF18OC0033926). M.O. acknowledges funding from the Instituto Nacional de Bioinformática, The EU BioExcel Centre of Excellence for HPC and the Spanish Ministry of Science (PID2021-122478NB-I00) and the Instituto de Salud Carlos III–Instituto Nacional de Bioinformatica (ISCIII PT 17/0009/0007 co-funded by the Fondo Europeo de Desarrollo Regional). M.O. is an ICREA Academy scholar and J.A. is a Juan de la Cierva fellow. M.C. was supported by institutional funds of the Max Planck Society. This project has been carried out using the resources of CSUC. IRB Barcelona is the recipient of a Severo Ochoa Award of Excellence from MINECO (Government of Spain).

## Author contributions

Conceptualization: A.E., J.G., R.C. and X.S. Experimental data acquisition, processing, and analysis: A.E., J.P., T.D., B.M., C.G., M.S.-N., M.B., L.S., J.G, and O.M. Simulations and data analysis: A.E., J.A., B.T. and R.C. Ensemble generation, and analysis: A.E., M.C. Supervision: A.E., B.B.K., O.M., M.O., R.C., and X.S. Writing original draft: A.E., J.G., R.C. and X.S. Writing final version: all authors. Funding acquisition: B.B.K., O.M., M.O., R.C., and X.S.

## Competing interests

M.B. and X.S. are founders of Nuage Therapeutics. M.B. is an employee of Nuage Therapeutics. X.S. is a scientific advisor of Nuage Therapeutics. The remaining authors declare no competing interests.
