## [Peer Review File · Nature Communications]

REVIEWER COMMENTS

Reviewer #1 (Remarks to the Author):

The manuscript by Salvatella and team describes the thorough investigation of a glutamine side chain-to-backbone interaction that stabilizes the alpha-helical fold. This is based on findings the group had reported previously (Nat. Commun. 10, 2034 (2019) and Biophys. J. 110, 2361–2366 (2016) – references 11 and 12). Here, it is put in a new sequence context and applied for the rational design of a stabilized helix. The concept intriguing with potential for application as it relies on natural amino acids only. However, to confirm the potential of this approach appropriate control experiments need to be performed. These are the main points (full list below): i) to validate the helix-stabilizing effect reference peptides are missing, and ii) in particular in the last part, orthogonal biophysical characterization methods should be applied.

Introduction:

a) The authors write: 'Peptides have some of the advantages of small molecules, such as their ease of synthesis and rather high bioavailability,...' It is not clear why peptides can be considered possessing a generally high bioavailability, considering that oral availability and cell-penetration is limited in particular for stabilized alpha helices. The authors should adjust their statement.

b) The introduction highlights side chain-to-side cross-linking to stabilized alpha-helices. However, N-caps and hydrogen bond-surrogates are missing and should be discussed as these are also relevant to the reported approach.

Design of Gln-based single alpha-helices:

c) Figure 1a should include the one-letter-code for amino acids so the sequence can be derived from that figure. Otherwise it is necessary to consult the SI too often.

d) The effects of the four arrangements (P1-5, P2-6, P3-7 and P5-9) are described qualitatively. The authors should also quantify the effects considering that differences in helicity at room temperature may not correlate with thermal stability (e.g. melting temperatures should be determined).

e) Control peptides should be included to highlight what the basic helicity of the chosen peptide sequence is.

f) The authors select the P3-7 motif for further studies as this was the most helical. How general is this behaviour – this maybe sequence dependent. Did the authors check this?

Structure of a Gln-based single alpha-helix:

g) In Figure 2a the sequences of the peptide should be provided.

h) The authors observe highest rigidity in the centre of (P3-7)₂ peptide. How can be verified that this is due to the central G²L interactions and not just the usual trend that helices show increased flexibility towards the termini?

i) The coordinates of the helix model derived by NMR have to be provided as a supporting information file.

j) The authors write: 'These peptides were obtained by mutating the fourth Leu residue...' Here, 'mutating' should be replaced by 'variation of'.

k) The authors perform a search for naturally occurring (P3-7)_n. In order support their their claim they should synthesize some analogues and verify the increased helicity. Otherwise, this is a consideration to put in the conclusion section.

Glu residues as pH-sensitive conformational switches

l) The 'pH-sensitive switching' of a related helix not stabilized by Glu residues should be investigated as a reference. This should also show some pH-sensitivity.

Design of Gln-based single α -helices that bind a globular target

m) The binding affinity should be determined using an orthogonal method (e.g. ITC, fluorescence polarization and/or SPR).

n) Effect on binding should be compared with alternative helix stabilization approach (e.g. stapling or hydrogen bond-surrogate).

Supporting information

o) HPLC traces of peptides should be included.

p) Atomic coordinates of discussed models should be provided.

Reviewer #2 (Remarks to the Author):

This is an excellent and very thorough piece of work related to the novel design of glutamine-based single α -helix scaffolds. The applications are obvious and this presents a new platform for targeting globular proteins.

I recommend acceptance as is. The work is of general interest, it is well done, and I really have absolutely no criticisms.

Reviewer #3 (Remarks to the Author):

The work by Escobedo and coworkers is an elegant combination of computational and NMR assays to characterize the helical nature of peptides involving glutamine residues. The results and discussion are adequately described.

My expertise is mostly on the computational side, and most of my comments are related to this part of the methodology.

The introduction makes a good point about peptides as intermediates between small molecules and large biologics, and about their potential to inhibit intracellular proteins. However, there is no mention about the difficulties of peptides crossing membranes or their fast degradation by proteases. Are the peptides described in this work more permeable than other peptides? I would expect that making stable helices would slow down degradation, have they confirmed this?

I don't follow the point of ML as illustrated by structures from the AlphaFold database. If there are some conditions in which the structures produce a structure, they are going to be biased towards that by virtue of making predictions with MSAs. What does AF predict in the absence of MSAs for these peptides? I think this would align better with the spirit of the work. (see Bioarxiv paper from the Ovchinnikov group regarding the "biophysical" energy function that AF learns).

I'm not familiar with the Agadir algorithm, but it seems similar to the Lifson-Roig model. Am I correct in understanding that it produces a partition function to predict helix content? And if so, is adding a new energy term to account for side chain interactions just modifying the partition function? I would like to know more details than given in the text: are these modifications for all side chains or only added for Glutamine residues that the authors are studying? What is the training set to reduce the

RMSDHel and how are the authors accounting for overfitting or transferring to the rest of systems under study?

The CoMAND approach seems interesting to produce ensembles that agree with data, rather than attempting to derive an ensemble from the data. There are many force fields used in different stages of the manuscript, which I find confusing: Charmm36m, Charm22*, Amber99SB-disp and DES-amber. I can understand the use of Amber99SB-disp and DES-amber as they supposedly describe order/disorder ensembles better. Why is the charmm36m introduced later and why is the charmm22* used after that? There should be some explanation about the choice and rationale for force fields. Otherwise it almost feels like using the ff that gives the right answer for each assay – which I'm sure is not what the authors have done. Would the inclusion of ff19SB make sense in this dataset? cMAP corrections in ff19sb were derived to reproduce helix nucleation properties according to the Lifson Roig model.

Is the biasing of trajectories in L3X simulations only done for creating the initial structures or are biases maintained during the simulation? The text in this section is not clear. Are these ensembles only used to select structures via the CoMand approach? If the systems are restrained: could conformations outside the restrained ensemble contribute to the NMR ensemble? And if they are not restrained, what is the agreement of predicted helicities from simulations/ff and NMR?

I'm interested in the Chemical shift perturbation formula used, weighting the Nitrogen shifts by a factor of 1/5. I have seen different values in the literature (e.g. 1/6) and wonder if some consensus/rationale is possible.

What is the bias for the WTE simulations? Is that the same as for L3X simulations?

Reviewer #1 (Remarks to the Author):

The manuscript by Salvatella and team describes the thorough investigation of a glutamine side chain-to-backbone interaction that stabilizes the alpha-helical fold. This is based on findings the group had reported previously (Nat. Commun. 10, 2034 (2019) and Biophys. J. 110, 2361–2366 (2016) – references 11 and 12). Here, it is put in a new sequence context and applied for the rational design of a stabilized helix. The concept is intriguing with potential for application as it relies on natural amino acids only. However, to confirm the potential of this approach appropriate control experiments need to be performed. These are the main points (full list below): i) to validate the helix-stabilizing effect reference peptides are missing, and ii) in particular in the last part, orthogonal biophysical characterization methods should be applied.

Introduction:

a) The authors write: 'Peptides have some of the advantages of small molecules, such as their ease of synthesis and rather high bioavailability,...' It is not clear why peptides can be considered possessing a generally high bioavailability, considering that oral availability and cell-penetration is limited in particular for stabilized alpha helices. The authors should adjust their statement.

We wrote that sentence because peptides, unlike antibodies, can in favorable cases cross the cell membrane but agree with Reviewer 1 that, as currently phrased, it needs to be corrected. We have replaced it with the following "Peptides have some of the advantages of small molecules, such as their ease of synthesis, and some of the advantages of antibodies, such as their relatively large size. Peptides, therefore, have at least in principle great potential as modulators of protein-protein interactions for pharmacological applications^{4,5}."

b) The introduction highlights side chain-to-side cross-linking to stabilized alpha-helices. However, N-caps and hydrogen bond-surrogates are missing and should be discussed as these are also relevant to the reported approach.

As pointed out by the Reviewer it is important to mention, in the introduction, that approaches other than stapling can be used to increase the helical content of synthetic linear peptides. We have modified the relevant paragraph, that now reads as follows: "Introducing non-natural amino acids that act as potent N-caps⁷ or substituting the $i+4 \rightarrow i$ hydrogen bonds stabilizing this secondary structure by covalent, and therefore permanent, surrogates⁸ can be used to achieve this goal. Alternatively, specific amino acids can be introduced at positions i and $i+3$, $i,i+4$ or $i,i+7$, that are close in space in α -helices, and linked by different means^{9,10} such as by peptide stapling^{11,12}. Peptide stapling is based ...".

Design of Gln-based single alpha-helices:

c) Figure 1a should include the one-letter-code for amino acids so the sequence can be derived from that figure. Otherwise it is necessary to consult the SI too often.

To address this concern Figure 1, and the rest of figures, have been modified to include the sequences. In addition we have included schematics representing the Gln side chain to main chain interactions stabilizing their helical conformation. We thank the Reviewer for this request as it has indeed led to an increase in clarity.

d) The effects of the four arrangements (P1-5, P2-6, P3-7 and P5-9) are described qualitatively. The authors should also quantify the effects considering that differences in helicity at room temperature may not correlate with thermal stability (e.g. melting temperatures should be determined).

We would like to highlight that the relationship between residue-specific helical propensity (p_{Hel}) and $^1\text{H}_\alpha$ and $^{13}\text{C}_\alpha$ chemical shifts measured by solution NMR is quantitative and, therefore, that the spectra represented in Figure 1b show, quantitatively and at residue-level, that the helical propensity of peptide P3-7 is higher than that of other peptides, in agreement with the CD data shown in Figure S1c. We agree with the reviewer that such differences in helical propensity at low or room temperature would not necessarily correlate with thermal stability but would like to clarify that the purpose of the experiment was to determine the former, as a first step towards the design of single, thermostable alpha helices by accumulating a larger number of such interactions: indeed peptide P3-7 contains two concatenated Gln side chain to main chain interactions whereas (P3-7)₂ contains four of them and (P3-7)₃, that we used for the applications reported in Figure 6, six of them. We have introduced changes in the presentation of these results to make this more clear: it now reads "... we used solution state nuclear magnetic resonance (NMR) spectroscopy to probe their structural properties by exploiting the quantitative dependence of $^{13}\text{C}_\alpha$ and $^1\text{H}_\alpha$ NMR chemical shifts on residue-specific helical propensity, where larger $^{13}\text{C}_\alpha$ and lower $^1\text{H}_\alpha$ shifts indicate higher helicity²³. An analysis of the NMR spectra indicated that in the sequence context of this family of peptides the strength of two $\text{Gln}_{i+4} \rightarrow \text{Leu}_i$ interactions is maximal when the donor of the first and the acceptor of the second interaction share a peptide bond such that the two interactions are concatenated, as in peptide P3-7; these results were confirmed by circular dichroism (CD) spectroscopy (Fig. S1c).".

e) Control peptides should be included to highlight what the basic helicity of the chosen peptide sequence is.

To address this we have included and discussed in the revised version the characterization of a control peptide, named (P3-7)_{3 Ctrlr}, where the six Leu amino acids accepting the side chain to main chain interactions have been substituted by Ala, that is a worse acceptor of such hydrogen bonds¹: the characterization is presented in Figure 1d,f. The results, that include CD spectra at 278 and 308K (Fig. 1d) and thermal unfolding curves (Fig. 1f), show that the helical content of the peptide decreases upon Leu substitution by Ala, despite Ala having high helical propensity, indicating that the side chain to main chain interactions contribute to helicity, as expected. In addition Fig. S1d now includes the SEC-MALS profile of (P3-7)_{3 Ctrlr} showing that it is monomeric. Finally we would like to point out that equivalent control experiments were carried out with the δ and γ peptides presented in Figure 6, where mutations of Leu to Ala (peptides δ_{Ctrl} and γ_{Ctrl}) led to decreases in helical content as well as in affinity for RAP74-CTD. We agree with the reviewer that it is appropriate to present evidence for this earlier in the paper and thank her/him for this suggestion.

f) The authors select the P3-7 motif for further studies as this was the most helical. How general is this behavior – this maybe sequence dependent. Did the authors check this?

We designed these peptides to have a sequence nucleating helix formation at the N-terminus (residues PGAS) and for the Gln side chain to main chain hydrogen bonds to form in an Ala-rich sequence context but, as suggested by the reviewer whether the arrangement of Gln side chain to main chain hydrogen bonds of peptide P3-7 maximizes helicity may to some extent depend on sequence context. We would like to point out, however, that in the sequence contexts of both peptides studied in Figure 6, derived from two different motifs interacting with RAP74-CTD, introduction of the arrangement of Gln side chain to main chain hydrogen bonds P3-7 led to clear increases in helical propensity in both cases. We have modified the sentence where these results are presented to acknowledge the point made by the reviewer as described in our response to point d).

Structure of a Gln-based single alpha-helix:

g) In Figure 2a the sequences of the peptide should be provided.

Please see above our response to point c).

h) The authors observe highest rigidity in the centre of (P3-7)₂ peptide. How can be verified that this is due to the central GL interactions and not just the usual trend that helices show increased flexibility towards the termini?

We agree with the Reviewer that helix fraying, due to the fact that the main chain to main chain hydrogen bonds stabilizing α -helices are necessarily unsatisfied in the first and last residues of linear peptides, can contribute to the decay of helicity that we observe in peptide (P3-7)₂, presented in detail in Figure 2. To make this clear to our readers we have added to the revised version a sentence in the description of the results presented in Figure 2a, that reads as follows "Although this result could be influenced by the well-characterized phenomenon of helix fraying²⁶ the main chain amide ¹⁵N relaxation data thus localizes the region of highest structural rigidity..".

i) The coordinates of the helix model driven by NMR have to be provided as a supporting information file.

The ensemble shown in Figure 2g is available as a .pdb file as supplementary material and also as PDB entry 8B1X.

j) The authors write: 'These peptides were obtained by mutating the fourth Leu residue...' Here, 'mutating' should be replaced by 'variation of'.

We have addressed this by replacing "mutating ... to" by "substituting ... by", as this is the way we have expressed this concept elsewhere in the manuscript. The relevant sentence now reads "These peptides were obtained by substituting the fourth Leu residue of peptide L₄Q₁₆, excised from AR, by 13 different representative amino acids (Fig. 3a)". We thank the Reviewer for pointing this out.

k) The authors perform a search for naturally occurring (P3-7)_n. In order support their their claim they should synthesize some analogues and verify the increased helicity. Otherwise, this is a consideration to put in the conclusion section.

We agree with the Reviewer that the prediction that the motifs identified are helical, in the absence of experimental validation, belongs to the discussion and have introduced this change in the revised version.

Glu residues as pH-sensitive conformational switches

l) The 'pH-sensitive switching' of a related helix not stabilized by Glu residues should be investigated as a reference. This should also show some pH-sensitivity.

This is indeed an important point that we failed to address in the original manuscript. As requested we have included to the revised version a study of the structural properties at pH 2.8 of peptides (P3-7)₂ and (P3-7)₃, that are stabilized by Gln (not Glu) side chain to main chain hydrogen bonds and represent appropriate control sequences. Figs. 4j and 4k show a comparison of the CD spectra of these peptides at pH 2.8 and 7.4 (and 278 K), as well as thermal denaturation curves. In agreement with our conclusions the helical propensity and thermal stability of these peptides is not altered by pH, indicating that the change in helical character in peptides E(P3-7)₂ and E(P3-7)₃ at pH 2.8 is indeed due to the re-protonation, under these conditions, of the Glu side chains. We have added a sentence to the relevant section to present these observations "; importantly, this increase of helicity and thermostability was not observed for peptides (P3-7)₂ and (P3-7)₃, that are stabilized by Gln side chain to main chain hydrogen bonds that are not affected by pH changes.". We thank the Reviewer for requesting that we present this important observation.

Design of Gln-based single α -helices that bind a globular target

m) The binding affinity should be determined using an orthogonal method (e.g. ITC, fluorescence polarization and/or SPR).

To address this point we studied the interaction between peptide δ and RAP74-CTD by ITC (Figure S11b). Despite the affinity being too low to be determined accurately by using this technique – the c parameter for an interaction with a $K_D > 100 \mu\text{M}$ falls outside the range leading to a reliable characterization of the thermodynamics of binding – the results are in semi-quantitative agreement with those obtained by NMR; to mention this result we added a sentence to the relevant section of the revised version "This interaction was also studied by isothermal titration calorimetry (ITC, Fig. S11b)." We also studied the interaction between peptide γ and RAP74-CTD but the affinity was too low to be measured reliably.

n) Effect on binding should be compared with alternative helix stabilization approach (e.g. stapling or hydrogen bond-surrogate).

The Reviewer makes an important point: to address it we have compared the affinities for RAP74-CTD of the peptides that we designed (δ and γ) to those of stapled peptides δ_{Stpl} and γ_{Stpl} as shown in Figure 6a,d and below:

```
 $\delta$  : KKPGASL AALQELQLLQLLQELQAAQAACKY
 $\delta_{\text{Stpl}}$ : KKPGASAAAAQEXQLAXLAQEAQAQAACKY

 $\gamma$  : KKPGASLSWLQLFQEEQAAQAACK
 $\gamma_{\text{Stpl}}$ : KKPGASASWXQLFXEEQAAQAACK
```

Figure 6a shows that peptide δ_{Stpl} has a helical propensity similar to that of peptide δ at 278 K, whereas at 298 K, the temperature at which the titration monitored by NMR was carried out, it is slightly higher than that of δ . Figures 6b, 6c and S11a show the results of the titrations, which report that the binding of peptide δ_{Stpl} is in the mM range, at least an order of magnitude weaker than the binding of δ . We hypothesize that the stronger binding of δ is due to an increased ability, relative to that of δ_{Stpl} , to adapt to the surface of RAP74-CTD.

Figure 6d shows that the helical propensity of peptide γ_{Stpl} is higher than that of peptide γ both at 278 and 298 K. Figures 6e, 6f and S11c show the results of the titrations, which report that the binding affinity of both γ and γ_{Stpl} for RAP74-CTD is in the mid-micromolar range. We hypothesize that the higher helicity we observed for γ_{Stpl} compensates its likely higher rigidity and is at least partly responsible for the 3-fold increase in the observed K_D .

We conclude that concatenating Gln side to main chain hydrogen bonds leads to enhancements in affinity that are in the same range as those obtained by stapling and that how well the two approaches compare depends on the specific sequence considered and the structure of the surface patch. We have added text throughout the relevant section to discuss these findings and our conclusions.

Supporting information

o) HPLC traces of peptides should be included.

Uncropped chromatograms of all peptides used in this study are provided in Figures S1d, S4a, S8a, S9 and S10a as obtained from size exclusion chromatography coupled to multiple angle light scattering (SEC-MALS). These were obtained on an Äkta Purifier FPLC system equipped with a Superdex Peptide 10/300GL size exclusion column. This technique was selected for the characterization of our peptide samples over HPLC-MS given its ability to report on the oligomeric state of our samples, which we find essential in our study. In unclear cases, we provide in addition mass characterization by native-MS.

p) Atomic coordinates of discussed models should be provided.

A spreadsheet is included as supplementary information that contains the Uniprot accession codes for all 2303 AlphaFold models discussed, along with the sequence and secondary structure analysis

of the (P3-7)_n-like motifs that they contain. In the same file, in different sheets, the models generated with Colabfold with and without MSAs are also shown. The .pdb files with the Colabfold-generated models are also available as supplementary material in a compressed .zip file.

Reviewer #2 (Remarks to the Author):

This is an excellent and very thorough piece of work related to the novel design of glutamine-based single α -helix scaffolds. The applications are obvious and this presents a new platform for targeting globular proteins.

I recommend acceptance as is. The work is of general interest, it is well done, and I really have absolutely no criticisms.

We thank the reviewer for such a positive and encouraging evaluation of our work.

Reviewer #3 (Remarks to the Author):

The work by Escobedo and coworkers is an elegant combination of computational and NMR assays to characterize the helical nature of peptides involving glutamine residues. The results and discussion are adequately described.

We thank the Reviewer for this positive assessment of our work.

My expertise is mostly on the computational side, and most of my comments are related to this part of the methodology.

The introduction makes a good point about peptides as intermediates between small molecules and large biologics, and about their potential to inhibit intracellular proteins. However, there is no mention about the difficulties of peptides crossing membranes or their fast degradation by proteases. Are the peptides described in this work more permeable than other peptides? I would expect that making stable helices would slow down degradation, have they confirmed this?

The Reviewer makes a fair point. We now include data relative to the stability of (P3-7)₃ and (P3-7)_{3 Ctrl} in human plasma, as well as evidence showing internalization of these two peptides in HeLa cells. Figure S2a shows that the plasma stability of both of them is atypically high for linear peptides^{2,3} with half-lives beyond 24 h. In order to show how this contrasts with values most often reported in the literature, we have also studied the peptide Angiopep-2 (TFFYGGSRGKRNNFKTEEY-OH), which displays a half-life of ca. 1h in agreement with values previously reported in the literature⁴. Figure S2b shows that both peptides are spontaneously internalized in HeLa cells, which is also uncommon for linear peptides of this size. These properties are likely related to the specific sequence of these peptides, which will change upon motif grafting for target binding (Fig. 6), possibly introducing protease cleavage sites. Yet, our results show that our scaffold is an excellent platform with advantageous pharmacokinetic properties for peptide drug development. The implications of these findings are now presented in the discussion: "We also showed that our scaffold peptides are highly resistant to proteolytic degradation in human plasma and readily internalized by living human cells (Fig. S2), thus overcoming some of the most important hurdles for peptide therapeutics. Although this could be related to their amino acid composition and sequence and needs to be studied on a case-by-case basis for potential hits, our design constitutes a valuable platform from which to evolve peptides with favorable pharmacokinetic properties."

I don't follow the point of ML as illustrated by structures from the AlphaFold database. If there are some conditions in which the structures produce a structure, they are going to be biased towards that by virtue of making predictions with MSAs. What does AF predict

in the absence of MSAs for these peptides? I think this would align better with the spirit of the work. (see Bioarxiv paper from the Ovchinnikov group regarding the “biophysical” energy function that AF learns).

We agree with Reviewer 3 that the MSAs could bias AlphaFold predictions towards folded structures. In fact polyQ tracts represent a clear example of this: AlphaFold predicts they form stable helices whereas experimental evidence, some of it from our own previous work, shows otherwise⁵. For this reason, and also given the suggestion of Reviewer 1, we have moved these findings to the discussion. We have also included two new results, that support conclusion, to the revised version:

- 1. We have updated our AlphaFold Database search to include models available since the July 2022 release. This has augmented the power of our analysis by increasing the total number of analysed structures from 50 to 2303 (Figure S7c). The main conclusion holds: naturally-occurring (P3-7)_n-like motifs tend to be helical in AFDB models, especially when helix breakers (Gly and Pro) are absent in the centre of the sequence.*
- 2. We now include a helical content analysis of (P3-7)_n-like motifs in a subset of 42 structural models calculated with and without MSAs using ColabFold⁶ (Figure S7c). A similar analysis performed on structural models of the same subset of proteins obtained from the AlphaFold Database is included for comparison. The results of the analysis on ColabFold-derived models calculated with MSAs are consistent with those obtained from the analysis of AFDB models. ColabFold-derived models calculated without MSAs show a reduction of roughly 10% of the natural (P3-7)_n-like motifs being helical, indicating that conditional MSA-derived training might play a role but also that AF energy function has learned that (P3-7)_n-like motifs have an intrinsic tendency to be helical.*

I’m not familiar with the Agadir algorithm, but it seems similar to the Lifson-Roig model. Am I correct in understanding that it produces a partition function to predict helix content? And if so, is adding a new energy term to account for side chain interactions just modifying the partition function? I would like to know more details than given in the text: are these modifications for all side chains or only added for Glutamine residues that the authors are studying? What is the training set to reduce the RMSD_{Hel} and how are the authors accounting for overfitting or transferring to the rest of systems under study?

The Reviewer is right: Agadir is indeed similar to the Lifson-Roig and Zimm-Bragg models. As explained by its original developers they differ “in several aspects of the statistical approximation and of the model for α -helix formation” but “[...] give virtually identical results when an equivalent set of parameters is used”. Agadir uses a partition function in which the statistical weight of helical conformations K is defined as:

$$K = e^{-\Delta G_{Hel}/RT}$$

where:

$$\Delta G_{Hel} = \Delta G_{Int} + \Delta G_{Hbond} + \Delta G_{SD} + \Delta G_{nonH}$$

where “ ΔG_{Int} is the summation of the intrinsic tendency of the j residues to adopt helical dihedral angles”, “ ΔG_{Hbond} is the sum of the net contribution of all the main chain-main chain hydrogen bonds within the helical region”, “ ΔG_{SD} is the sum of the net contribution of all the side chain-side chain interactions located at positions $i,i+3$ and $i,i+4$ within the helical region, with respect to the random-coil state, plus weakly attractive and repulsive coulombic interactions [...]”, and “ ΔG_{nonH} is the sum of the net contribution to the stability of the helical region of all the residues that are not in the helical conformation”.

Agadir’s “energy contributions are empirically obtained from previous experimental studies in designed peptides and site-directed mutagenized proteins.” Such energy contributions are tabulated in the different categories indicated above, so that for example the original reference shows all possible $i,i+3$ and $i,i+4$ interactions energy terms between all natural amino acid types in

Table 2. Although these are in principle thought to account for side chain - side chain interactions, and side chain to main chain hydrogen bonds are not explicitly considered, with the help of Prof. Luis Serrano (Centre for Genomic Regulation, Barcelona) we systematically modified the $L_{i,i+4}$ interaction energy term obtaining the results shown in Figures S1a and S1b. Therefore, the introduction of the energy term we describe here does slightly modify the partition function just for the $L_{i,i+4}$ interaction.

The results we present here show that the value of our introduced energy term increases with length, which supports the notion that side chain to main chain hydrogen bonds act cooperatively to stabilize the polyQ helix: the longer the polyQ, the more interactions reinforcing each other, the stronger the energy term. We find that by simply comparing Agadir's predicted helicity with our measurements for increasingly long polyQ tracts, for increasing values of our energy term, from which we calculate the $RMSD_{Hel}$ shown in Figs. S1a and S1b. Due to the cooperative nature of this unparameterized interaction, and the consequent variability of the value of the new energy term, we did not attempt to re-train Agadir to include it, so we did neither generate a training set nor evaluate the model's overfit or transferability.

To ensure this is now clear to our readers we have added the following to the description of Agadir in the Results section: "The helicity of peptides and intrinsically disordered (ID) proteins can be predicted by secondary structure prediction algorithms such as Agadir. This algorithm, similarly to the Zimm-Bragg¹⁹ and Lifson-Roig²⁰ helix-coil transition models, is based on statistical mechanics²¹. The statistical weight of the helical state of any peptide segment depends on the free energy of it folding into an α -helix, that Agadir computes as the sum of different terms including one accounting for interactions between residues at positions i and $i+4$, that require experimental parametrization. The current version of Agadir does not account for $Gln_{i+4} \rightarrow Leu_i$ side chain to main chain interactions stabilizing polyQ helices and therefore underestimates the helicity of the polyQ tract in AR: peptide L_4Q_{16} , excised from AR and harboring four such interactions, has 38% helical propensity according to NMR experiments while Agadir predicts only 3%¹⁴. To address this we introduced an additional term to the free energy of folding into an α -helix accounting for this interaction ($\Delta G_{i,i+4}^{LQ}$) and by minimizing the prediction error with respect to the NMR derived helicity ($RMSD_{Hel}$, see supplementary methods) obtained $\Delta G_{i,i+4}^{LQ} = -0.6 \text{ kcal mol}^{-1}$ for L_4Q_{16} , in the range expected for one hydrogen bond in water²²."

The CoMAND approach seems interesting to produce ensembles that agree with data, rather than attempting to derive an ensemble from the data. There are many force fields used in different stages of the manuscript, which I find confusing: Charmm36m, Charm22*, Amber99SB-disp and DES-amber. I can understand the use of Amber99SB-disp and DES-amber as they supposedly describe order/disorder ensembles better. Why is the charmm36m introduced later and why is the carmm22* used after that? There should be some explanation about the choice and rational for force fields. Otherwise it almost feels like using the ff that gives the right answer for each assay – which I'm sure is not what the authors have done. Would the inclusion of ff19SB make sense in this dataset? cMAP corrections in ff19sb were derived to reproduce helix nucleation properties according to the Lifson Roig model.

We understand the concern of the Reviewer and acknowledge that a better explanation regarding the choices of force field throughout the manuscript, that we have now added to the relevant parts of the Methods section, was necessary. As stated by the Reviewer a99sb-disp and DES-amber are indeed considered to be some of the best-performing force fields for describing both structured and unstructured proteins. In the case of $(P3-7)_2$ structure determination (Figure 2), we required unrestrained frame pools that properly reflect the experimental helix-coil equilibrium to yield sufficient realistic frames for the CoMAND approach. For this reason, we calculated 200 ns WTE simulations with a99sb-disp and DES-amber. Following the suggestion of the Reviewer, we have now calculated a 200 ns WTE trajectory of this peptide using the ff19sb force field, which showed a

similar helicity profile. We thank the Reviewer for his suggestion and will consider the usage of ff19sb in ensemble generation in the future.

The information we required from the simulations of the L_3XQ_{16} peptides was different. In that case we were not intending to generate ensembles that reproduce the experimental helicity data per se; instead, we needed good estimates of the effect that rotamer sampling of the side chain of residue X_i has on the solvation of the $Q_{i+4} H_{\epsilon 21}$ donor. To best show the robustness of our conclusions, we decided to use two force fields of different origins: whereas a99sb-disp and DES-amber are both Amber force fields with similar side chain parametrization, Charmm36m offered the possibility of an alternative side chain parametrization. Thus, we chose a99sb-disp and Charmm36m to perform simulations on segments of the L_3XQ_{16} peptides that were restrained to keep them helical and favor the formation of the $Q_{i+4} \rightarrow X_i$ side chain to main chain hydrogen bond (see below): this was necessary as unrestrained simulations did not provide sufficient sampling of the relevant conformations (Figures S5 and S6, see below). Figure S7a shows the side chain rotameric sampling for the X_i side chain in both a99sb-disp and Charmm36m in comparison with that of the relevant residues in helical conformations in the PDB. Thus, the L_3XQ_{16} simulations were merely used to obtain estimates of the $Q_{i+4} H_{\epsilon 21}$ donor SASA which, in independent combination with six different helical propensity scales, are good predictors of the experimental data regardless of which force field we used to obtain them, even considering their different origin (Figure 3f).

Finally, we used Charmm22* to generate the seeds and treat the classical subsystem in our QM/MM simulations to enable the comparison of electron density and polarization in $E_{i+4} \rightarrow L_i$ vs $Q_{i+4} \rightarrow L_i$ side chain to main chain hydrogen bonds, since this was the force field we used to derive the glutamine data in our previous related publication¹. Thus, the data shown in Figures 4h and 4i is directly comparable and we can conclude that $E_{i+4} \rightarrow L_i$ h-bonds are stronger than $Q_{i+4} \rightarrow L_i$ ones.

Is the biasing of trajectories in L3X simulations only done for creating the initial structures or are biases maintained during the simulation? The text in this section is not clear. Are these ensembles only used to select structures via the CoMand approach? If the systems are restrained: could conformations outside the restrained ensemble contribute to the NMR ensemble? And if they are not restrained, what is the agreement of predicted helicities from simulations/ff and NMR?

We have now clarified this in the main text (Methods section) and with the addition of two new supplementary figures (S5 and S6). The latter show additional data on 133 preparatory simulations.

As explained above, the L_3XQ_{16} simulations were not produced to generate ensembles that reproduce the experimental data, but rather to obtain good estimates of the donor $H_{\epsilon 21}$ SASA.

Relevant conformations to obtain such values involve the $X_i - Q_{i+4}$ segment folded as a helix and with a sufficient sampling of the side chain to main chain interaction, which NMR relaxation data shows is transient (Figure 2a). Classical non-polarizable force fields are not particularly parameterized for side chain to main chain hydrogen bonds, so we introduced two restraints to sample them sufficiently: one backbone restraint that keeps the peptide folded as a helix most of the time and one soft side chain restraint that keeps the $Q_{i+4} H_{\epsilon 21}$ donor and the $X_i O$ acceptor in close proximity to favor the establishment of the interaction while respecting its transient nature. The spring constant k for both restraints was optimized so that a minimum value was used to ensure we obtained a wide dynamic range of $H_{\epsilon 21}$ SASA values. The restraints were maintained across the simulations.

Figures S5a and S6a (42 x 500 ns trajectories each) show the effect of increasing the value of k to restrain the backbone angles Φ and Ψ to -60 and -40 degrees, respectively. For both a99sb and Charmm36m we selected a value for k of $5 \text{ kJ mol}^{-1} \text{ degree}^{-2}$, which generates trajectories whose back-calculated chemical shifts are closest to the experimental data, ensuring sufficient frames with a helical fold.

Figures S5b and S6b (21 and 28 x 1 μs trajectories respectively) show the effect of increasing the value of k to restrain the distance between the $Q_{i+4} H_{\epsilon 21}$ donor and the $X_i O$ acceptor in the 0 to 4 \AA range. As anticipated, not restraining results in the uneven sampling of the interaction across simulations: compare for example simulations for L3Y and L3T with all others in Fig. S5 top left, which shows the $Q_{i+4} H_{\epsilon 21} - X_i O$ distance distributions for simulations with no side chain restraint ($k = 0 \text{ kJ mol}^{-1} \text{ nm}^{-2}$). This is because the donor side chain was free to sample the entire rotameric space, and the sampling of the relevant interaction was uneven in the trajectories of different peptides. Instead, the introduction of a soft restraint ensured comparability across simulations and a wide dynamic range of the donor $H_{\epsilon 21}$ SASA values (see shadowed panels in Figs. S5b and S6b). Instead, an excessively strong spring constant k value did not allow for the interaction to behave transiently and resulted in a poor dynamic range in SASA estimation (Figs S5b bottom right, Fig S6b bottom left). The value of k was independently optimised for each force field (50 and 250 $\text{kJ mol}^{-1} \text{ nm}^{-2}$ for Charmm36m and a99sb-disp, respectively), accounting for their different side chain parametrization.

I'm interested in the Chemical shift perturbation formula used, weighting the Nitrogen shifts by a factor of 1/5. I have seen different values in the literature (e.g. 1/6) and wonder if some consensus/ rationale is possible.

Different equations (sometimes a simple sum of the weighted absolute shift changes) and different scaling factors have been described in the literature for averaging chemical shift perturbations. Values of the scaling of the ^{15}N chemical shifts between 0.1⁸ and 0.45^{9,10} have been used, with many examples in between. Some authors have used the relationship between gyromagnetic ratios: 0.102¹¹. Others have used the chemical shift range for amide ^1H and ^{15}N nuclei to calculate a scaling factor of 0.14¹². However, depending on the ^1H and ^{15}N chemical shift dispersions, each protein could yield a different value. Another option has been to determine the scaling factor from the ratio of the average variances of the amide ^{15}N and ^1H chemical shifts observed for all amino acid residues in proteins available in the BMRB database. This method provided values ranging from 0.15 to 0.2¹³ The performance of the different ways of combining chemical shifts has been examined systematically concluding that the differences between methods are small and there seems not to be an ideal or theoretically justifiable weighting factor^{14,15}.

What is the bias for the WTE simulations? Is that the same as for L3X simulations?

Just to clarify, only the a99sb-disp and DES-amber simulations used to generate the CoMAND frame pool for $(P3-7)_2$ structure determination were performed in the PT-WTE ensemble. L_3XQ_{16} simulations were not because the restraints explained above were used. For the simulations performed in the PT-WTE ensemble, details regarding the bias used have been added to the

methods section: "During the bias-deposition, a Gaussian with height 1.2 kJ mol^{-1} and a width of 140 kJ mol^{-1} was added every 0.5 ps . The bias-factor was set to 16".

References

1. Escobedo, A. *et al.* Side chain to main chain hydrogen bonds stabilize a polyglutamine helix in a transcription factor. *Nat. Commun.* **10**, 2034 (2019).
2. Cardle, I. I., Jensen, M. C., Pun, S. H. & Sellers, D. L. Optimized serum stability and specificity of an $\alpha\text{v}\beta\text{6}$ integrin-binding peptide for tumor targeting. *J. Biol. Chem.* **296**, 100657 (2021).
3. Howell, S. M. *et al.* Serum stable natural peptides designed by mRNA display. *Sci. Rep.* **4**, 6008 (2014).
4. Wei, X. *et al.* Retro-inverso isomer of Angiopep-2: a stable d-peptide ligand inspires brain-targeted drug delivery. *Mol. Pharm.* **11**, 3261–3268 (2014).
5. Reid Alderson, T., Pritišanac, I., Moses, A. M. & Forman-Kay, J. D. Systematic identification of conditionally folded intrinsically disordered regions by AlphaFold2. *bioRxiv* 2022.02.18.481080 (2022) doi:10.1101/2022.02.18.481080.
6. Mirdita, M. *et al.* ColabFold: making protein folding accessible to all. *Nat. Methods* **19**, 679–682 (2022).
7. Muñoz, V. & Serrano, L. Elucidating the folding problem of helical peptides using empirical parameters. *Nat. Struct. Biol.* **1**, 399–409 (1994).
8. Allain, F. H. *et al.* Solution structure of the HMG protein NHP6A and its interaction with DNA reveals the structural determinants for non-sequence-specific binding. *EMBO J.* **18**, 2563–2579 (1999).
9. Burnett, R. *et al.* Global estimates of mortality associated with long-term exposure to outdoor fine particulate matter. *Proceedings of the National Academy of Sciences* **115**, 9592–9597 (2018).
10. Coggins, B. E. *et al.* Structure of the LpxC deacetylase with a bound substrate-analog inhibitor. *Nat. Struct. Biol.* **10**, 645–651 (2003).
11. Geyer, M., Herrmann, C., Wohlgemuth, S., Wittinghofer, A. & Kalbitzer, H. R. Structure of the Ras-binding domain of RalGEF and implications for Ras binding and signalling. *Nat. Struct. Biol.* **4**, 694–699 (1997).
12. Williamson, R. A., Carr, M. D., Frenkiel, T. A., Feeney, J. & Freedman, R. B. Mapping the binding site for matrix metalloproteinase on the N-terminal domain of the tissue inhibitor of metalloproteinases-2 by NMR chemical shift perturbation. *Biochemistry* **36**, 13882–13889

(1997).

13. Mulder, F. A., Schipper, D., Bott, R. & Boelens, R. Altered flexibility in the substrate-binding site of related native and engineered high-alkaline *Bacillus subtilis*ins. *J. Mol. Biol.* **292**, 111–123 (1999).
14. Schumann, F. H. *et al.* Combined chemical shift changes and amino acid specific chemical shift mapping of protein-protein interactions. *J. Biomol. NMR* **39**, 275–289 (2007).
15. Williamson, M. P. Using chemical shift perturbation to characterise ligand binding. *Prog. Nucl. Magn. Reson. Spectrosc.* **73**, 1–16 (2013).

REVIEWERS' COMMENTS

Reviewer #1 (Remarks to the Author):

The revised version of the manuscript addresses all points raised by the reviewers and I recommend publication as is.

Reviewer #3 (Remarks to the Author):

The authors have addressed my concerns.